# SLC7A11 expression level dictates differential responses to oxidative stress in cancer cells

Yuelong Yan [1], Hongqi Teng[1], Qinglei Hang [1], Lavanya Kondiparthi[2,5], Guang Lei [1], Amber Horbath [1], Xiaoguang Liu [1], Chao Mao[1], Shiqi Wu[1], Li Zhuang[1], M. James You[3], Masha V. Poyurovsky[2], Li Ma [1,4], Kellen Olszewski[2,6] & Boyi Gan [1,4] ✉

The cystine transporter solute carrier family 7 member 11 (SLC7A11; also called xCT) protects cancer cells from oxidative stress and is overexpressed in many cancers. Here we report a surprising finding that, whereas moderate over-expression of SLC7A11 is beneficial for cancer cells treated with $H_2O_2$, a common oxidative stress inducer, its high overexpression dramatically increases $H_2O_2$-induced cell death. Mechanistically, high cystine uptake in cancer cells with high overexpression of SLC7A11 in combination with $H_2O_2$ treatment results in toxic buildup of intracellular cystine and other disulfide molecules, NADPH depletion, redox system collapse, and rapid cell death (likely disulfidptosis). We further show that high overexpression of SLC7A11 promotes tumor growth but suppresses tumor metastasis, likely because metastasizing cancer cells with high expression of SLC7A11 are particularly susceptible to oxidative stress. Our findings reveal that SLC7A11 expression level dictates cancer cells' sensitivity to oxidative stress and suggests a context-dependent role for SLC7A11 in tumor biology.

Cysteine is a proteinogenic amino acid that plays a pivotal role in maintaining cellular redox homeostasis by providing the rate-limiting precursor for synthesizing glutathione (GSH) and other biomolecules involved in antioxidant defense[1,2]. Cancer cells, particularly metastasizing cancer cells, often experience high levels of oxidative stress and thus have a high demand for cysteine from the extracellular space to build up their antioxidant defense systems[3–6]. However, because of the oxidizing extracellular environment, extracellular cysteine is unstable and quickly converts to cystine, the oxidizing dimeric form of cysteine, whose concentration in the extracellular space is typically higher than that of cysteine by an order of magnitude[7,8]. Therefore, most cancer cells mainly rely on the cystine transporter solute carrier family 7 member 11 (SLC7A11; also called xCT) to import extracellular cystine[9]. Once imported into the cytosol, cystine is reduced to cysteine, which is

subsequently used to synthesize GSH for antioxidant defense[9,10]. SLC7A11 has a well-established role in protecting cells from oxidative stress–induced cell death (e.g., ferroptosis), and it is overexpressed in many human cancers[8].

However, high cystine uptake also represents a vulnerability in SLC7A11-overexpressing cancer cells[11]. Cystine is the least soluble of the common amino acids, and its aberrant accumulation in the cytosol can be highly cytotoxic because it induces disulfide stress, forcing cells to quickly reduce cystine to cysteine, which is much more soluble[11]. Since the reduction of cystine to cysteine consumes nicotinamide adenine dinucleotide phosphate (NADPH), a reducing equivalent, SLC7A11-overexpressing cells with high rates of cystine import require large amounts of NADPH to constantly reduce cystine to cysteine and to maintain intracellular cystine at nontoxic levels. Consequently, such

[1]Department of Experimental Radiation Oncology, The University of Texas MD Anderson Cancer Center, Houston, TX 77030, USA. [2]Kadmon Corporation, LLC (A Sanofi Company), New York, NY 10016, USA. [3]Department of Hematopathology, The University of Texas MD Anderson Cancer Center, Houston, TX 77030, USA. [4]The University of Texas MD Anderson Cancer Center UTHealth Graduate School of Biomedical Sciences, Houston, TX 77030, USA. [5]Present address: Sanofi US Services Inc, 270 Albany St, Cambridge, MA 02139, USA. [6]Present address: The Barer Institute, Philadelphia, PA 19104, USA. ✉e-mail: bgan@mdanderson.org

cells exhibit marked accumulation of intracellular cystine and other disulfide molecules and undergo rapid cell death termed disulfidptosis in response to glucose starvation; under these conditions, NADPH generation through the pentose phosphate pathway and other routes is significantly impaired[12–14]. However, it remains unclear whether disulfide stress and the resulting cell death in SLC7A11-overexpressing cells only occur under glucose starvation or whether they also occur under other NADPH-depleting conditions. Addressing this question will help understand the metabolic nature of disulfide stress–induced cell death in SLC7A11-overexpressing cancer cells and identify therapeutic strategies to target SLC7A11-overexpressing cancers.

As noted above, SLC7A11-mediated cystine transport plays an essential role in suppressing ferroptosis, a form of iron-dependent, regulated cell death triggered by unchecked lipid peroxidation in cellular membranes[8]. Indeed, ferroptosis was discovered from studies understanding the potent cell-killing effect of erastin, a SLC7A11 inhibitor[15]. Recent studies have established ferroptosis as a key tumor-suppression mechanism and have shown that SLC7A11 overexpression in cancer cells promotes tumor growth at least partly by suppressing ferroptosis[8,16–18]. However, SLC7A11's role in cell death promotion in glucose-starved cells also indicates that its role in tumor biology is complex and likely context-dependent, although the exact context(s) in which SLC7A11 might have a tumor-suppressive function remains unknown.

In this study, we showed that, whereas moderate overexpression of SLC7A11 protects cancer cells from $H_2O_2$-induced cell death, counterintuitively, high overexpression of SLC7A11 potently induces cell death under $H_2O_2$ treatment by depleting NADPH and inducing disulfide stress. We further showed that high overexpression of SLC7A11 promotes primary tumor growth but suppresses tumor metastasis, likely because metastasizing cancer cells with high overexpression of SLC7A11 are particularly vulnerable to oxidative stress. Our findings suggest that SLC7A11 expression levels dictate its context-dependent roles in mediating cancer cells' oxidative stress responses and in tumor biology.

## Results

### Moderate and high overexpression of SLC7A11 exert opposite effects on $H_2O_2$-induced cell death

We sought to understand whether SLC7A11 overexpression promotes cell death under NADPH-depleting conditions other than glucose starvation. $H_2O_2$-induced oxidative stress can be neutralized by antioxidant enzymes such as glutathione peroxidase 1 (GPX1), which uses GSH as its co-factor to reduce $H_2O_2$ to $H_2O$; at the same time, GSH is oxidized to glutathione disulfide (GSSG), which is subsequently converted back to GSH by glutathione reductase (GR) at the expense of NADPH (Supplementary Fig. 1a). Consequently, $H_2O_2$ treatment decreases intracellular NADPH reserves[5,6]. However, SLC7A11 has a well-established role in protecting cells from $H_2O_2$-induced oxidative stress and cell death[8]. We reasoned that SLC7A11-mediated cystine uptake might have a dual role in redox maintenance in the context of $H_2O_2$ treatment. On one hand, the reduction of cystine to cysteine consumes NADPH, which would exacerbate the NADPH-depleting effects caused by $H_2O_2$ treatment, thereby promoting disulfide stress. On the other hand, cysteine provides the key precursor for synthesizing GSH to counteract $H_2O_2$-induced oxidative stress (Supplementary Fig. 1a). Accordingly, we reasoned that SLC7A11 expression levels might dictate its potential pro- or anti-cell death effect under $H_2O_2$ treatment.

To test this hypothesis, we analyzed SLC7A11 expression and cystine uptake levels across a panel of cancer cell lines. These analyses showed that cystine uptake levels generally correlated with expression levels of SLC7A11 (but not with those of other proteins involved in cystine uptake, such as SLC7A9 and SLC3A1) in these cell lines (Fig. 1a, b). We further categorized these cell lines into SLC7A11-low,

-moderate, and -high cells based on their relative SLC7A11 expression and cystine uptake levels, in which SLC7A11-moderate (UMRC6, H226, A498, and A549) and -high cell lines (T98G and Hs578T) exhibited 2-5– and 10-fold increases, respectively, in cystine uptake compared to SLC7A11-low cell lines (H1299 and 786-O).

We then reduced SLC7A11 expression to different levels in SLC7A11-high T98G cells. We found that depleting SLC7A11 by the CRISPR-Cas9 approach reduced cystine uptake by ~90% (Fig. 1c) and, as expected, sensitized T98G cells to $H_2O_2$-induced cell death (Fig. 1d). Surprisingly, knocking down *SLC7A11* using short hairpin RNAs (shRNAs) with a moderate reduction in cystine uptake (Fig. 1e) reduced cell death in $H_2O_2$-treated cells (Fig. 1f). We confirmed these findings in another SLC7A11-high cell line, Hs578T cells (Fig. 1g–j). In contrast, knocking down *SLC7A11* in SLC7A11-moderate cell lines (A498, UMRC6, H226, and A549) increased $H_2O_2$-induced cell death (Supplementary Fig. 1b, c).

Conversely, we induced the overexpression of SLC7A11 at different levels in SLC7A11-low H1299 cells (Fig. 1a, k). Cystine uptake measurements showed that moderate or high overexpression of SLC7A11 (SLC7A11-moderate or -high) increased cystine uptake by about 5- and 9-fold, respectively (Fig. 1k), which corresponded to cystine uptake levels in cell lines with moderate or high expression of endogenous SLC7A11 (Fig. 1a). (Of note, we had to overexpress SLC7A11 in SLC7A11-low cells to a higher level than that of endogenous SLC7A11 in SLC7A11-moderate cell lines in order to achieve a corresponding moderate increase of cystine uptake. SLC7A11 requires the chaperone protein SLC3A2 for its localization on the plasma membrane to mediate cystine uptake, and SLC3A2 also binds to several other amino acid transporters[19]; consequently, we need to overexpress SLC7A11 at higher levels to compete with other transporters for partnering with SLC3A2 in cells.) Notably, while moderate overexpression of SLC7A11 suppressed $H_2O_2$-induced cell death, as expected, high overexpression of SLC7A11 dramatically promoted cell death under $H_2O_2$ treatment (Fig. 1l). We made similar observations in another SLC7A11-low cell line 786-O cells with moderate or high overexpression of SLC7A11 (Fig. 1a, m, n).

Together, our data suggest that moderate and high expression of SLC7A11 have opposite effects—protective and sensitizing effects, respectively—on $H_2O_2$-induced cell death. In contrast, both moderate and high overexpression of SLC7A11 promoted cell death in glucose-starved H1299 and 786-O cells (Fig. 1l, n), whereas both *SLC7A11* knockout and knockdown protected T98G and Hs578T cells from glucose starvation–induced cell death; in addition, *SLC7A11* knockout had much more pronounced protective effects against glucose starvation–induced cell death than did *SLC7A11* knockdown in these SLC7A11-high cells (Fig. 1d, f, h, j), which is consistent with the findings of previous reports[11,13,20–22]. Therefore, the peculiar effects caused by differential expression of SLC7A11 apply to only $H_2O_2$-treated cells.

We next examined whether the cell death induced in $H_2O_2$-treated SLC7A11-high cells can be rescued by any known cell-death inhibitor. We found that different cell death inhibitors, including the apoptosis inhibitor Z-VAD-FMK, the ferroptosis inhibitor ferrostatin-1, and the necroptosis inhibitor necrostatin-1s, showed no rescuing effect on $H_2O_2$-induced cell death in SLC7A11-high cancer cells, including 786-O cells with high overexpression of SLC7A11 and T98G cells (Supplementary Fig. 1d–h). We confirmed the rescuing effects of Z-VAD-FMK on staurosporine-induced apoptosis and of ferrostatin-1 on RSL3-induced ferroptosis in these cells (Supplementary Fig. 1d–h). Since the rescuing effect of Z-VAD-FMK on staurosporine-induced apoptosis was partial in T98G cells (Supplementary Fig. 1e), as a complementary approach, we also deleted the genes *BAX* and *BAK*, which are essential to apoptosis induction, in these SLC7A11-high cell lines using the CRISPR/Cas9 approach (Supplementary Fig. 1i). Our results confirmed that *BAX/BAK* genetic ablation abolished staurosporine-induced apoptosis (Supplementary Fig. 1j) but did not affect the cell death triggered by $H_2O_2$ treatment (Supplementary Fig. 1k). Finally, we

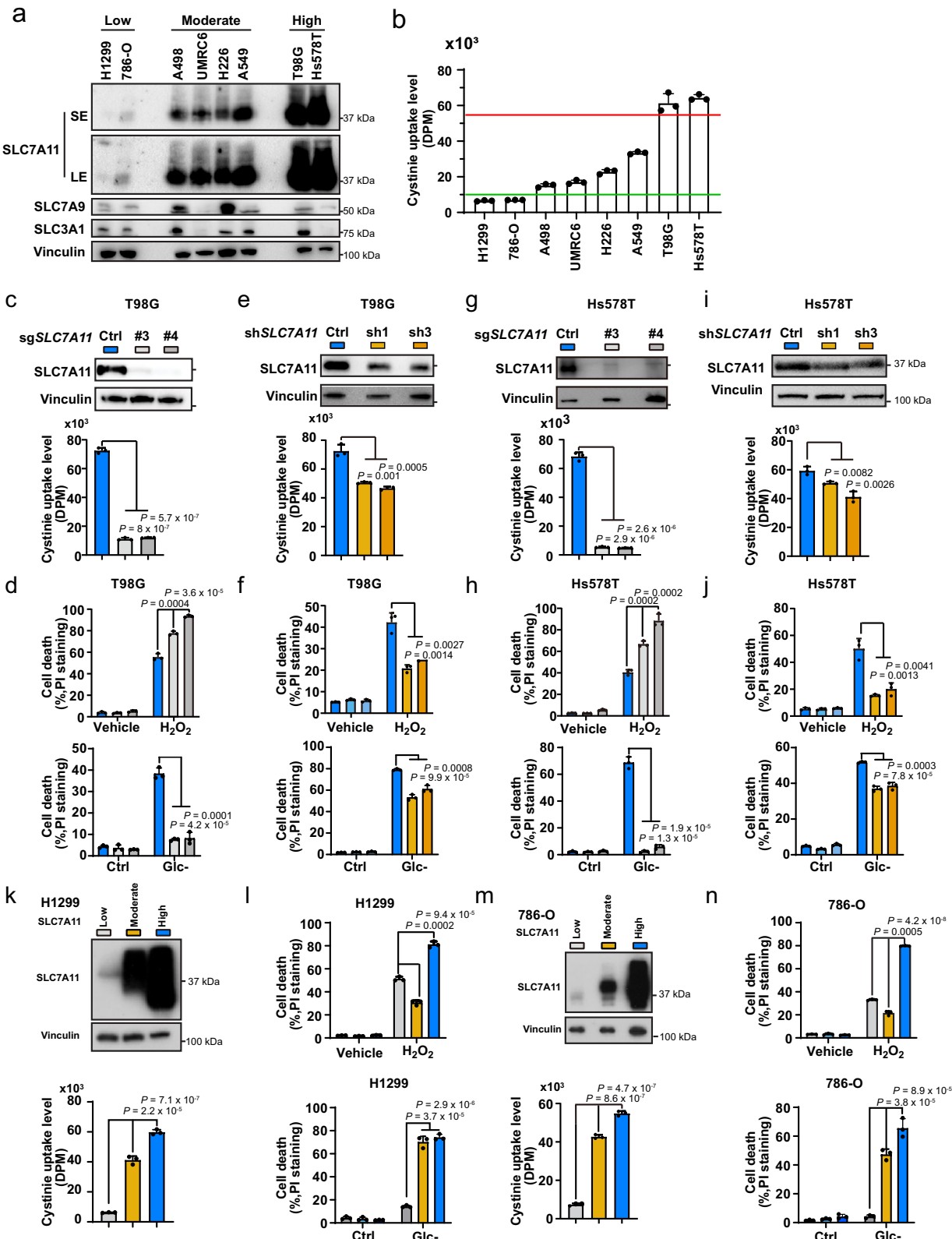

showed that $H_2O_2$-induced cell death in SLC7A11-low counterparts could be partially rescued by apoptosis and necroptosis inhibitors (Supplementary Fig. 1l), which is consistent with a recent report[23]. These data suggest that, while $H_2O_2$ mainly induces apoptosis and necroptosis in SLC7A11-low cells, $H_2O_2$ induces a different form of cell death in SLC7A11-high cells (which is not apoptosis, ferroptosis, or necroptosis). Our following data suggest that this cell death most likely

is the recently described disulfidptosis[14] (see also "Discussion" section).

### High SLC7A11 expression induces disulfide accumulation in $H_2O_2$-treated cells

Previously, we showed that glucose starvation in cancer cells with moderate or high overexpression of SLC7A11 resulted in the marked

**Fig. 1 | Moderate and high overexpression of *SLC7A11* have opposite effects on H2O2-induced cell death. a** Protein levels of SLC7A11, SLC7A9, and SLC3A1 in a panel of cancer cell lines. Vinculin was used as the loading control. **b** Cystine uptake levels in a panel of cancer cell lines. **c** Protein levels of SLC7A11 (up) and cystine uptake levels (down) in sgCtrl and *SLC7A11* knockout T98G cells. **d** Cell death in response to 1 mM H2O2 treatment (up) or glucose starvation (down) in T98G cells with indicated genotypes for 24 h was measured using PI staining. **e** Protein levels of SLC7A11 (up) and cystine uptake levels (down) in shCtrl and *SLC7A11* knockdown T98G cells. **f** Cell death in response to 1 mM H2O2 treatment (up) or glucose starvation (down) in T98G cells with indicated genotypes for 24 h was measured using PI staining. **g** Protein levels of SLC7A11 (up) and cystine uptake levels (down) in sgCtrl and *SLC7A11* knockout Hs578T cells. **h** Cell death in response to 1 mM H2O2 treatment (up) or glucose starvation (down) in Hs578T cells with indicated genotypes for 24 h was measured using PI staining. **i** Protein levels of SLC7A11 (up) and cystine uptake levels (down) in shCtrl and *SLC7A11* knockdown Hs578T cells. **j** Cell death in response to 1 mM H2O2 treatment (up) or glucose starvation (down) in Hs578T cells with indicated genotypes for 24 h was measured using PI staining. **k** Protein levels of SLC7A11 (up) and cystine uptake levels (down) in SLC7A11-low, -moderate, and -high H1299 cells. **l** Cell death in response to 1 mM H2O2 treatment (left) or glucose starvation (right) for 20 h in SLC7A11-low, -moderate, and -high H1299 cells was measured using PI staining. **m** Protein levels of SLC7A11 (up) and corresponding cystine uptake levels (down) in SLC7A11-low, -moderate, and -high 786-O cells. **n** Cell death in response to 1 mM H2O2 treatment (left) or glucose starvation (right) in SLC7A11-low, -moderate, and -high 786-O cells for 20 h measured using PI staining. Data were presented as mean ± SD; *n* = 3. *n* indicates independent repeats. *P* value was determined by two-tailed unpaired Student's *t* test. Source data are provided as a Source Data file.

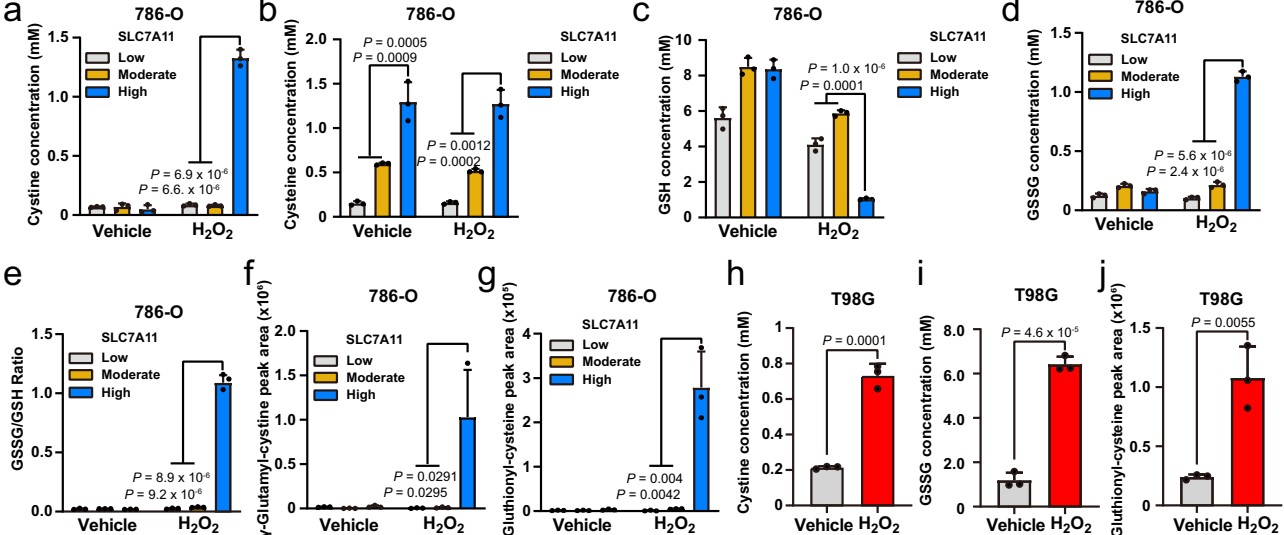

**Fig. 2 | High SLC7A11 expression leads to marked increases in intracellular disulfide molecules under H2O2 treatment. a–e** Measurement of intracellular concentrations of cystine (**a**), cysteine (**b**), GSH (**c**), GSSG (**d**), and the GSSG/GSH ratio (**e**) in SLC7A11-low, -moderate, and -high 786-O cells treated with vehicle or 1 mM H2O2 for 2 h. **f, g** Relative levels of intracellular γ-glutamyl-cystine (**f**) and γ-glutathionyl-cysteine (**g**) in SLC7A11-low, -moderate, and -high 786-O cells treated with vehicle or 1 mM H2O2 for 2 h. **h, i** Measurement of intracellular concentrations of cystine (**h**) and GSSG (**i**) in T98G cells treated with vehicle or 1 mM H2O2 for 8 h. **j** Relative levels of intracellular γ-glutathionyl-cysteine in T98G cells treated with vehicle or 1 mM H2O2 for 8 h. Data were presented as mean ± SD; *n* = 3. *n* indicates independent repeats. *P* value was determined by two-tailed unpaired Student's *t* test. Source data are provided as a Source Data file.

accumulation of intracellular cystine and other disulfide molecules as well as NADPH depletion, and rapid cell death[12]. We therefore measured the levels of intracellular cysteine, cystine, and other disulfide molecules in SLC7A11-low, -moderate, and -high 786-O cells (i.e., parental 786-O cells and isogenic derivatives with moderate or high overexpression of SLC7A11) treated with vehicle or H2O2. Once imported into cells through SLC7A11, cystine is quickly reduced to cysteine in the cytosol[11,12]. Consistent with this, in SLC7A11-low, -moderate, and -high cells treated with vehicle, we observed no obvious differences in intracellular cystine levels, but intracellular cysteine levels increased with the level of SLC7A11 overexpression across cell types (Fig. 2a, b). Remarkably, H2O2 treatment drastically increased intracellular cystine levels in SLC7A11-high (but not in SLC7A11-moderate) cells compared to SLC7A11-low cells (Fig. 2a). Of note, a dramatic increase in cystine levels was not accompanied by a corresponding decrease in cysteine levels in SLC7A11-high cells under H2O2 treatment (Fig. 2b). We previously made similar observations in glucose-starved SLC7A11-high cells[12]. This is likely because these stress conditions—H2O2 treatment and glucose starvation—suppress downstream cysteine-consuming processes such as protein synthesis; consequently, intracellular cysteine levels remain relatively stable despite steep rises in intracellular cystine levels in SLC7A11-high cells under H2O2 treatment. These observations suggest that cysteine level change

is unlikely to be the mechanism causing increased cell death in these contexts.

Further analyses revealed that SLC7A11-high (but not SLC7A11-moderate) cells exhibited GSH depletion (Fig. 2c) but marked increases in the levels of other disulfide molecules such as γ-glutamyl-cystine and γ-glutathionyl-cysteine (the cysteinyl disulfides of γ-glutamyl-cysteine and GSH, respectively; Supplementary Fig. 2), and GSSG; the GSSG/GSH ratio also increased in H2O2-treated cells (Fig. 2d–g). Likewise, we observed that H2O2 treatment increased the levels of intracellular cystine, GSSG, and γ-glutathionyl-cysteine in T98G cells (Fig. 2h–j), although the fold change was not as pronounced as that in 786-O cells with high overexpression of SLC7A11 (Fig. 2a–g). Finally, consistent with a recent report[24], we found that H2O2 treatment increased cystine uptake in SLC7A11-low, -moderate, and -high cells, leading to an even more pronounced increase of cystine uptake in SLC7A11-high cells compared to SLC7A11-moderate counterparts under H2O2 treatment (Supplementary Fig. 2c), which might contribute to the substantially increased intracellular cystine level in H2O2-treated SLC7A11-high cells. Collectively, our results showed that, under H2O2 treatment, high (but not moderate) overexpression of SLC7A11 results in accumulation of intracellular disulfides, which correlates with increased susceptibility to H2O2-induced cell death.

## Inhibiting SLC7A11-mediated cystine uptake and resolving disulfide accumulation prevent high SLC7A11−induced cell death under H₂O₂ treatment

We next sought to determine whether SLC7A11-mediated cystine uptake and disulfide accumulation underlie high SLC7A11−induced cell death under $H_2O_2$ treatment. To this end, we took several approaches to preventing disulfide accumulation. These included (i) treatment with the SLC7A11 inhibitor erastin to block SLC7A11-mediated cystine uptake (Supplementary Fig. 3a, b); (ii) treatment with the disulfide-reducing agents tris(2-carboxyethyl) phosphine (TCEP) and 2-mercaptoethanol (2-ME) to reduce cystine to cysteine in the medium (and thereby bypass SLC7A11-mediated cystine transport) (Supplementary Fig. 3c); (iii) and treatment with N-acetyl cysteine (NAC) and penicillamine (including both D- and L-penicillamine) to regenerate free thiols via disulfide exchange (i.e., Cys-Cys + NAC → Cys + NAC-Cys; Cys-Cys + penicillamine → Cys + penicillamine-Cys; Supplementary Fig. 3d). These treatments partially or completely abolished disulfide accumulation in $H_2O_2$-treated 786-O cells with high SLC7A11 overexpression (Fig. 3a–e) and, correspondingly, markedly suppressed $H_2O_2$-induced cell death in these cells (Fig. 3f–k). Likewise, these treatments suppressed $H_2O_2$-induced cell death in SLC7A11-high T98G cells (Fig. 3l–q). We confirmed that erastin treatment potently suppressed cystine uptake (Supplementary Fig. 3e) and that treatment with TCEP or 2-ME increased extracellular cysteine levels (Supplementary Fig. 3f). (Of note, the effect of 2-ME on increasing extracellular cysteine levels was more moderate than that of TCEP. This is because the mechanistic bases for how TCEP and 2-ME reduce extracellular cystine are somewhat different: TCEP directly cleaves cystine to two molecules of cysteine, whereas 2-ME reacts with cystine to generate cysteine and a mixed disulfide of 2-ME and cysteine [i.e., Cys-Cys + 2-ME → Cys + 2-ME-Cys]. Because extracellular cysteine is unstable, it continues to react with another molecule of 2-ME [or a 2-ME-2-ME disulfide], ending up with most products as 2-ME-Cys disulfide, which can be taken up into cells via system L[25] and is subsequently converted back to cysteine inside cells, thereby still bypassing SLC7A11-mediated cystine transport to provide intracellular cysteine for GSH synthesis (Supplementary Fig. 3c)).

$H_2O_2$ is known to induce reactive oxygen species (ROS)[5,6]. We therefore measured ROS levels in vehicle- or $H_2O_2$-treated 786-O cells with low, moderate, or high SLC7A11 expression. Our results showed that, under normal culturing conditions, SLC7A11-moderate or -high cells exhibited lower levels of ROS than did SLC7A11-low cells and that, as expected, $H_2O_2$ treatment increased ROS levels (Supplementary Fig. 3g). Under $H_2O_2$ treatment, moderate overexpression of SLC7A11 suppressed ROS levels, whereas high overexpression markedly increased them (Supplementary Fig. 3g). We then examined whether increased ROS levels play any causal role in $H_2O_2$-induced cell death in SLC7A11-high cells. We found that treatment with the ROS scavenger Trolox potently suppressed $H_2O_2$-induced ROS (Supplementary Fig. 3h, i), but did not affect $H_2O_2$-triggered cell death in SLC7A11-high cancer cells (Supplementary Fig. 3j, k). Conversely, erastin treatment dramatically suppressed $H_2O_2$-induced cell death (Fig. 3f, l), but did not suppress $H_2O_2$-induced ROS (Supplementary Fig. 3l, m). These data therefore established an uncoupling between $H_2O_2$-induced cell death and ROS in SLC7A11-high cells. Collectively, our data suggest that cell death in SLC7A11-high cancer cells under $H_2O_2$ treatment is likely caused by SLC7A11-mediated cystine transport and disulfide accumulation, but not by ROS per se.

Finally, since SLC7A11 is an antiporter that imports cystine and exports glutamate, it is possible that the decreased intracellular glutamate levels (caused by SLC7A11-mediated glutamate export) might also contribute to $H_2O_2$-induced cell death in SLC7A11-high cells. Intracellular glutamate is mainly generated from glutamine through glutaminase (GLS; Supplementary Fig. 4a). Therefore, this hypothesis would predict that GLS inhibition, similar to high expression of

SLC7A11, should decrease intracellular glutamate level and promote $H_2O_2$-induced cell death, and that further depleting intracellular glutamate levels by combining GLS inhibition with SLC7A11 high expression should exacerbate $H_2O_2$-induced cell death. However, we found that, in SLC7A11-low cells, treatment with the GLS inhibitor CB-839 caused more decreases in intracellular glutamate levels than did high SLC7A11 overexpression (Supplementary Fig. 4b) yet failed to promote $H_2O_2$-induced cell death (Supplementary Fig. 4c). Furthermore, while CB-839 treatment further decreased intracellular glutamate levels in SLC7A11-high cells (Supplementary Fig. 4b), CB-839 even attenuated $H_2O_2$-induced cell death in SLC7A11-high cells (Supplementary Fig. 4c), likely because decreasing intracellular glutamate levels by CB-839 treatment suppressed SLC7A11-mediated cystine uptake (Supplementary Fig. 4d). Together, these data suggest that the increased cell death in $H_2O_2$-treated SLC7A11-high cells is not directly related to glutamate export.

## NADPH consumption contributes to increased sensitivity to H₂O₂ in SLC7A11-high cancer cells

Both the reduction of cystine to cysteine and antioxidant defenses against $H_2O_2$ require NADPH as the reducing equivalent (Supplementary Fig. 1a). We therefore measured NADPH levels in SLC7A11-low, -moderate, and -high cells during $H_2O_2$ treatment. As shown in Fig. 4a, $H_2O_2$ treatment increased the $NADP^+$/NADPH ratio much more substantially in SLC7A11-high cells than in SLC7A11-low or -moderate counterparts, indicating that NADPH is rapidly depleted in SLC7A11-high cells under $H_2O_2$ treatment. $H_2O_2$ treatment also increased the $NADP^+$/NADPH ratio in T98G cells and knocking down *SLC7A11* normalized the increased $NADP^+$/NADPH ratio under $H_2O_2$ treatment (Fig. 4b). Likewise, inhibiting SLC7A11-mediated cystine transport by erastin or sulfasalazine at least partly normalized the increased $NADP^+$/NADPH ratio caused by $H_2O_2$ treatment in these SLC7A11-high cells (Fig. 4c–e), suggesting that high rates of cystine uptake in SLC7A11-high cells contribute to NADPH depletion under $H_2O_2$ treatment.

Antioxidant defenses against $H_2O_2$ is mainly mediated by GSH-dependent peroxidases, such as GPX1. We therefore studied the role of GPX1 in $H_2O_2$-induced cell death in SLC7A11-high cells. We found that *GPX1* deletion in SLC7A11-low 786-O cells increased $H_2O_2$-induced cell death (which is consistent with the protective role of GPX1 in $H_2O_2$-induced oxidative stress; Fig. 4f, g); interestingly, *GPX1* deletion in SLC7A11-high counterparts had an opposite effect and moderately suppressed $H_2O_2$-induced cell death (Fig. 4f, g). Consistently, *GPX1* deletion also alleviated NADPH depletion and GSSG level (as well as the GSSG/GSH ratio) increases in $H_2O_2$-treated SLC7A11-high cells (Fig. 4h–j). It should be noted that the cell death reduction caused by *GPX1* deletion in $H_2O_2$-treated SLC7A11-high cells was moderate (Fig. 4g), likely because *GPX1* deletion on one hand reserves more NADPH and therefore protects SLC7A11-high cells from disulfide stress−induced cell death, but on the other hand still renders these cells susceptible to $H_2O_2$-induced oxidative stress, eventually resulting in cell death.

To further study the role of NADPH consumption in the $H_2O_2$-induced death of SLC7A11-high cells, we examined whether increasing NADPH consumption can sensitize SLC7A11-low cells to $H_2O_2$-induced cell death. To this end, we induced the expression of a genetically encoded, water-forming NADPH oxidase, triphosphopyridine nucleotide oxidase (TPNOX, which oxidizes cytosolic NADPH and therefore can be used as a genetic tool to increase the $NADP^+$/NADPH ratio[26]) in either SLC7A11-low or-high cancer cells (Fig. 4k). Under $H_2O_2$ treatment, TPNOX overexpression increased the $NADP^+$/NADPH ratio in SLC7A11-low cells but not in SLC7A11-high cells (Fig. 4l), likely because the SLC7A11-high cells already exhibited high $NADP^+$/NADPH ratios under these conditions. Correspondingly, TPNOX overexpression significantly increased cell death in $H_2O_2$-treated SLC7A11-low cells,

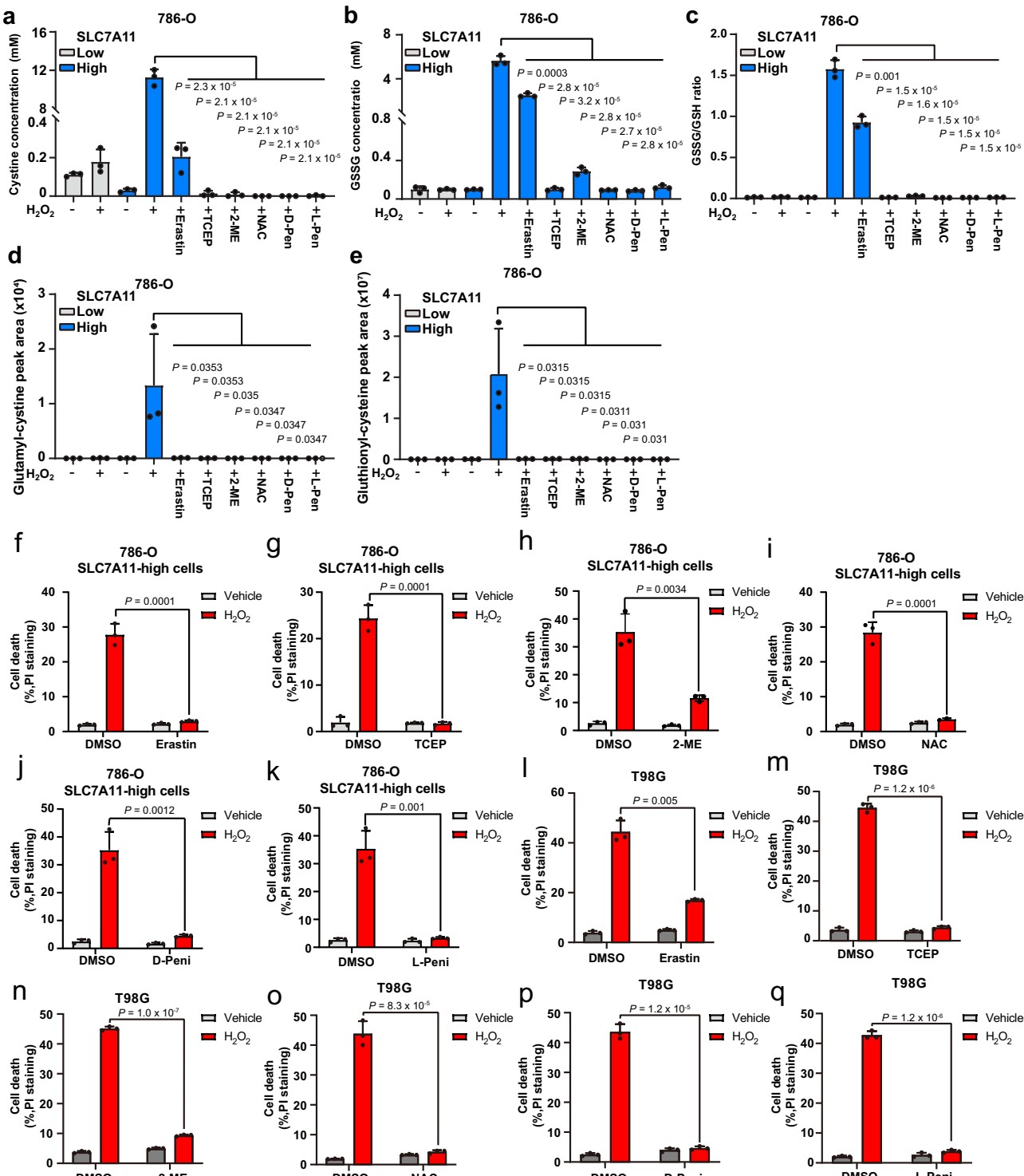

**Fig. 3 | Blocking cystine uptake and resolving disulfide stress suppress high SLC7A11−induced cell death under H₂O₂ treatment. a–c** Measurement of intracellular concentrations of cystine (**a**), GSSG (**b**), and the GSSG/GSH ratio (**c**) in SLC7A11-low 786-O cells treated with vehicle or 1 mM H₂O₂ for 2 h and SLC7A11-high 786-O cells treated with vehicle or 1 mM H₂O₂ for 2 h with or without 5 μM erastin, 1 mM tris(2-carboxyethyl) phosphine (TCEP), 1 mM 2-mercaptoethanol (2-ME), 2 mM *N*-acetyl cysteine (NAC), 2 mM ᴅ-penicillamine (D-Pen), or 2 mM ʟ- penicillamine (L-Pen). **d, e** Relative levels of intracellular γ-glutamyl-cystine (**d**) and γ-glutathionyl-cysteine (**e**) in SLC7A11-low 786-O cells treated with vehicle or 1 mM H₂O₂ for 2 h and SLC7A11-high 786-O cells treated with vehicle or 1 mM H₂O₂ for 2 h

with or without 5 μM erastin, 1 mM TCEP, 1 mM 2-ME, 2 mM ᴅ-penicillamine, or 2 mM ʟ-penicillamine. **f–k** Cell death in response to 1 mM H₂O₂ treatment for 6 h with or without 5 μM erastin (**f**), 1 mM TCEP (**g**), 1 mM 2-ME (**h**), 2 mM NAC (**i**), 2 mM ᴅ-penicillamine (**j**), or 2 mM ʟ-penicillamine (**k**) in SLC7A11-high 786-O cells measured using propidium iodide (PI) staining. **l–q** Cell death in response to 1 mM H₂O₂ treatment for 24 h with or without 5 μM erastin (**l**), 1 mM TCEP (**m**), 1 mM 2-ME (**n**), 2 mM NAC (**o**), 2 mM ᴅ-penicillamine (**p**), or 2 mM ʟ-penicillamine (**q**) in T98G cells measured using PI staining. Data were presented as mean ± SD; *n* = 3. *n* indicates independent repeats. *P* value was determined by two-tailed unpaired Student's *t* test. Source data are provided as a Source Data file.

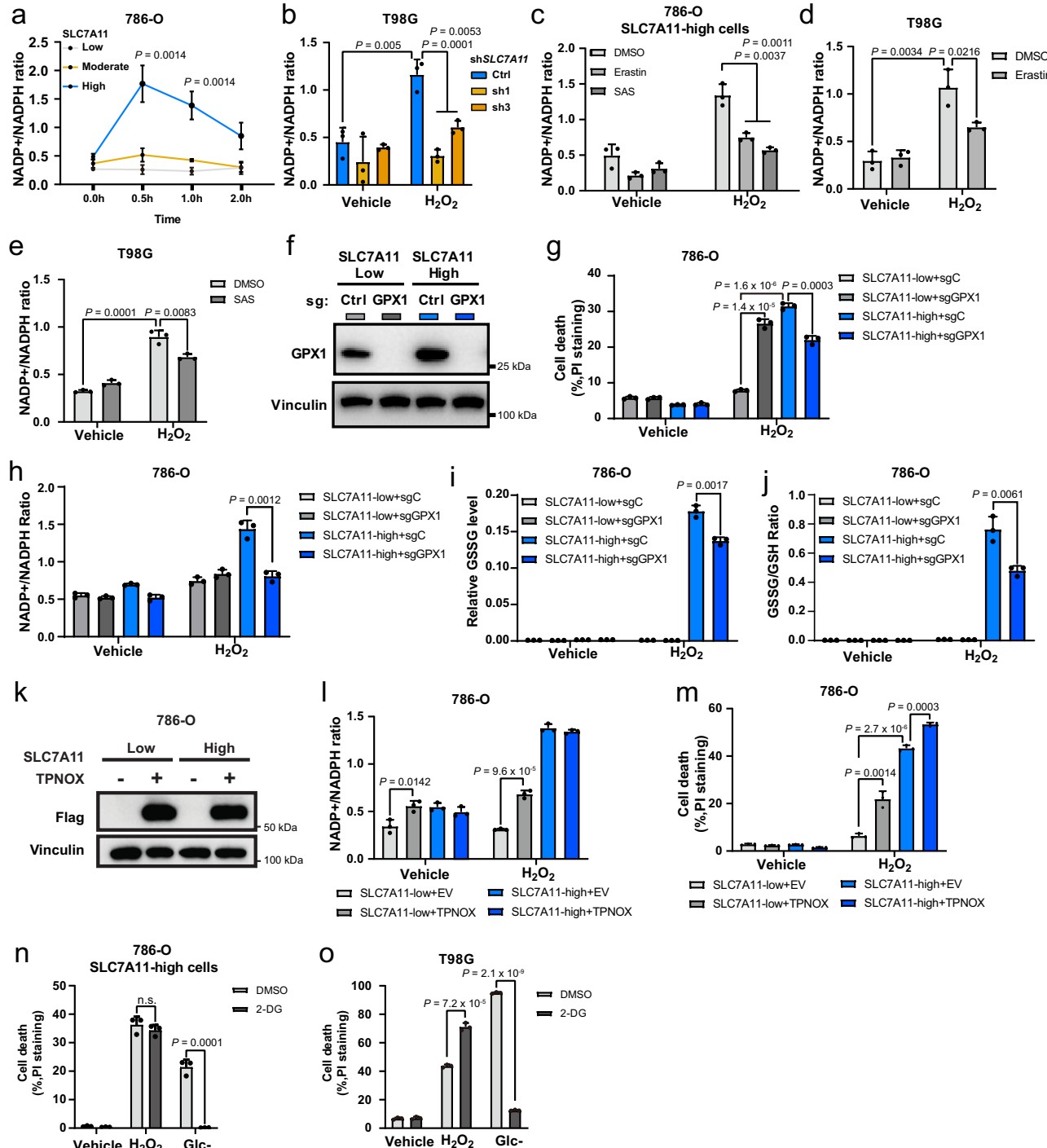

**Fig. 4 | NADPH consumption contributes to the increased sensitivity to H₂O₂ in SLC7A11-high cancer cells. a** Measurement of the NADP⁺/NADPH ratio in SLC7A11-low, -moderate, and -high 786-O cells treated with vehicle or 1 mM H₂O₂ at the indicated time points. **b** Measurement of the NADP⁺/NADPH ratio in shCtrl and *SLC7A11* knockdown T98G cells treated with vehicle or 1 mM H₂O₂ for 2 h. **c** Measurement of the NADP⁺/NADPH ratio in SLC7A11-high 786-O cells co-treated 1 mM H₂O₂ and 5 μM erastin or 10 μM sulfasalazine. **d**, **e** Measurement of the NADP⁺/NADPH ratio in T98G cells co-treated with 1 mM H₂O₂ and 5 μM erastin (**d**) or 10 μM sulfasalazine (SAS) (**e**). **f** Protein levels of GPX1 in SLC7A11-low and -high 786-O cells were measured using western blotting. Vinculin was used as the loading control. **g** Cell death in response to treatment with 1 mM H₂O₂ for 6 h in sgCtrl or sg*GPX1* infected SLC7A11-low and -high 786-O cells measured using PI staining. **h**–**j** Measurement of the NADP + /NADPH ratio (**h**), relative GSSG level (**i**), and GSSG/GSH ratio (**j**) in sgCtrl or *sgGPX1* infected SLC7A11-low and -high 786-O cells.

**k** Protein levels of Flag-triphosphopyridine nucleotide oxidase (TPNOX) in SLC7A11-low and -high 786-O cells. Vinculin was used as the loading control. **l** Measurement of the NADP⁺/NADPH ratio in SLC7A11-low and -high 786-O cells with empty vector (EV) or TPNOX overexpression under vehicle or 1 mM H₂O₂ treatment. **m** Cell death in response to treatment with 1 mM H₂O₂ for 6 h in SLC7A11-low and -high 786-O cells with EV or TPNOX overexpression measured using PI staining. **n** Cell death in response to treatment with 1 mM H₂O₂ treatment or glucose (Glc-) starvation for 6 h with or without 2 mM 2-deoxyglucose (2-DG) in SLC7A11-high 786-O cells measured using PI staining. **o** Cell death in response to treatment with 1 mM H₂O₂ or glucose starvation for 24 h with or without 2 mM 2-DG in T98G cells measured using PI staining. Data were presented as mean ± SD; *n* = 3. *n* indicates independent repeats. *P* value was determined by two-tailed unpaired Student's *t* test. n.s. not significant. Source data are provided as a Source Data file.

but its promotion of cell death in $H_2O_2$-treated SLC7A11-high cells was very moderate (Fig. 4m).

We previously showed that increasing cells' NADPH supply by 2-deoxyglucose (2-DG) treatment could prevent disulfide stress and cell death in glucose-starved, SLC7A11-overexpressing cancer cells (as a glucose analog, 2-DG can be shunted into the pentose phosphate pathway to generate NADPH)[12]. In the current study, we confirmed that 2-DG treatment prevented glucose starvation–induced cell death in 786-O cells with high overexpression of SLC7A11 and T98G cells; however, 2-DG treatment did not prevent $H_2O_2$-induced cell death in these SLC7A11-high cell lines (Fig. 4n, o). This result was not surprising. Under glucose starvation, 2-DG replaced glucose to supply NADPH to cells, whereas under $H_2O_2$ treatment, cells were cultured in a glucose-replete medium (under which condition glucose continues to support NADPH generation), which explains 2-DG's inability to prevent $H_2O_2$-induced cell death in these cells. These results also highlight the different mechanisms underlying NADPH depletion in SLC7A11-high cells under $H_2O_2$ treatment versus glucose starvation (see later for more detailed discussion). Together, our data suggest that high expression of SLC7A11 in combination with $H_2O_2$ treatment depletes NADPH, which potentially contributes to $H_2O_2$-induced cell death.

## High overexpression of SLC7A11 promotes primary tumor growth but suppresses tumor metastasis

The oxidizing environment in the blood contributes to the highly inefficient nature of metastasis. Most cancer cells die of oxidative stress during metastasis, and the few cancer cells that successfully metastasize often exhibit increased antioxidant capabilities, at least partly by upregulating NADPH-generating pathways[4,27]. Considering the high relevance of NADPH to metastasis, we examined the effects of low, moderate, or high SLC7A11 expression on metastasis (as well as on primary tumor growth). In 786-O xenograft tumor models, SLC7A11-high tumors exhibited significantly increased tumor growth compared with SLC7A11-moderate or -low tumors (Fig. 5a), which was consistent with our and others' previous findings revealing a role of SLC7A11 in promoting tumor growth[8,16,17,28].

We then injected the same amount of luciferase-labeled 786-O cells with high, moderate, and low expression levels of SLC7A11 intracardially into mice (Fig. 5b). Thirty minutes after intracardiac injection, we observed equal whole-body bioluminescence signals in the three groups of mice (Supplementary Fig. 5a; day 0), showing that the tumor cells had rapidly distributed throughout the mice's bodies. Weekly bioluminescent imaging showed that, 5 weeks after the injection, mice injected with 786-O cells with low or moderate SLC7A11 expression developed metastases to different organs, most notably the liver (Fig. 5c, d). Strikingly, although high overexpression of SLC7A11 promoted primary tumor growth (Fig. 5a), it significantly suppressed metastasis (Fig. 5c, d). Further analyses at the end point revealed that livers from mice injected with SLC7A11-low or -moderate cells exhibited much larger nodules than those from mice injected with SLC7A11-high cells (Fig. 5e). Histopathologic analyses showed that liver specimens from the mice injected with SLC7A11-low or -moderate cancer cells frequently harbored large metastatic nodules, whereas those injected with SLC7A11-high cancer cells occasionally contained small metastatic nodules (Fig. 5f). Immunohistochemical analyses confirmed stronger SLC7A11 staining in SLC7A11-high metastatic nodules than in SLC7A11-moderate or -low nodules (Fig. 5f), but there was no significant difference in Ki67 staining among the SLC7A11-high, -moderate, and -low metastatic nodules (Supplementary Fig. 5b, c); furthermore, SLC7A11-high modules showed decreased cleaved caspase-3 staining compared to SLC7A11-moderate or -low nodules (Supplementary Fig. 5b, d), which could not explain the reduced metastasis of SLC7A11-high tumors (Fig. 5d).

We further showed that NAC treatment almost completely restored metastasis in SLC7A11-high tumors to the level similar to that in vehicle-treated SLC7A11-low tumors, whereas Trolox has a minimal rescuing effect on SLC7A11-tumors (Fig. 5g), which is consistent with our in vitro data (Fig. 3i, o and Supplementary Fig. 3j, k) and suggests the cell death–promoting effect is at least partly responsible for high SLC7A11-mediated metastasis suppression. Conversely, TPNOX overexpression moderately decreased metastasis in SLC7A11-low tumors (Supplementary Fig. 5e), which is consistent with the relatively moderate effect of TPNOX overexpression on depleting NADPH and promoting $H_2O_2$-induced cell death than did high overexpression of SLC7A11 (see Fig. 4l, m). Finally, we showed that, in H1299 tumor models, high (but not moderate) overexpression of SLC7A11 promoted tumor growth (Supplementary Fig. 5f) yet suppressed metastasis (Fig. 5h–j and Supplementary Fig. 5g).

Our data from animal studies suggested that the oxidizing environment in the blood should select against circulating tumor cells (CTCs) with high SLC7A11 expression. To test this hypothesis, we examined *SLC7A11* expression levels from an RNA-Seq dataset with 16 CTCs and 12 available primary breast tumor samples from matched patients[29]. In support of our hypothesis, the analysis showed high SLC7A11 expression in 50% (6 of 12) primary tumor samples but only 12% (2 of 16) of the CTCs (Fig. 5k). Because the sample size in this study was limited, we further compared *SLC7A11* expression between primary breast tumor samples from The Cancer Genome Atlas dataset and breast CTCs from other studies[30,31] (with the caveat that these datasets were not generated from matched patients). The analysis again showed that primary breast tumors exhibited significantly higher SLC7A11 expression than did breast CTCs (Fig. 5l). Together, our data indicate that, at least in the cell lines or tumor models we have examined, high SLC7A11 overexpression promotes primary tumor growth but suppresses metastasis, likely because SLC7A11-high cancer cells are susceptible to cell death induced by oxidative stress during metastasis.

## Discussion

Previously, we and others showed that, in SLC7A11-overexpressing cancer cells, the reduction of cystine to cysteine consumes large amounts of NADPH. Consequently, SLC7A11 overexpression in combination with glucose starvation depletes intracellular NADPH and triggers disulfide stress and potent cell death[12,13]. These findings raised the question of whether SLC7A11 overexpression promotes disulfide stress and cell death under other NADPH-depleting conditions. In this study, we revealed that high overexpression of SLC7A11 also promotes NADPH depletion and triggers disulfide stress and subsequent cell death under $H_2O_2$ treatment, an oxidative stress–inducing condition known to consume NADPH. Therefore, high SLC7A11–induced cell death is not restricted to glucose starvation and can be extended to other NADPH-depleting conditions, highlighting that this type of cell death can occur under broader contexts.

We recently showed that excessive accumulation of intracellular disulfides in SLC7A11-moderate/-high cells under glucose starvation promotes aberrant disulfide bonding in actin cytoskeleton proteins, F-actin collapse, and cell contraction, and subsequently induces a distinctive form of cell death, which we termed disulfidptosis[14]. $H_2O_2$-induced cell death in SLC7A11-high cells share several features with disulfidptosis, including (1) cells under both conditions exhibit aberrant levels of intracellular disulfide molecules (such as cystine); (2) cell death can be prevented by disulfide-reducing agents (such as TCEP and 2-ME) or agents that regenerate free thiols via disulfide exchange (such as D- and L-penicillamine); and (3) cell death cannot be rescued by ROS scavengers (such as Trolox) or inhibitors that block other forms of cell death (such as apoptosis and ferroptosis). Therefore, we propose that $H_2O_2$-induced cell death in SLC7A11-high cells mostly likely is dislulfidptosis, although more study is required to further characterize this cell death.

Although both glucose starvation–induced and $H_2O_2$-induced cell death in SLC7A11-high cells are triggered by NADPH depletion and

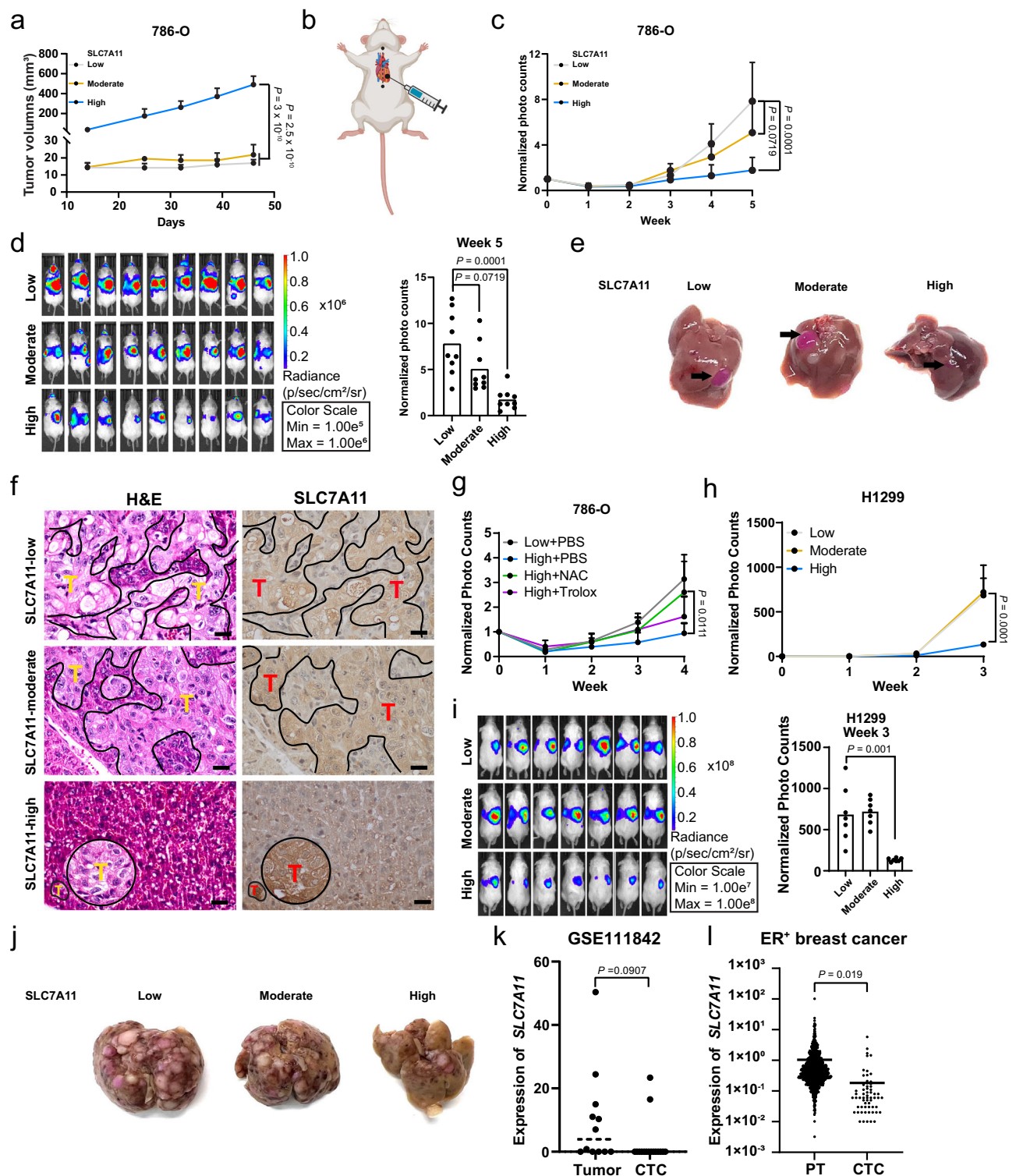

disulfide stress, the underlying mechanisms by which NADPH is depleted are somewhat different between these two conditions. Under glucose starvation, NADPH depletion is caused by the combined effect of blocking NADPH generation (by glucose deprivation) and increasing NADPH consumption (by enhancing cystine reduction in SLC7A11-overexpressing cells). In contrast, under $H_2O_2$ treatment, NADPH is depleted by the promotion of two NADPH-consuming processes—$H_2O_2$ treatment and cystine reduction—without the impairment of NADPH generation. Consequently, higher SLC7A11 expression is needed to deplete NADPH below the cell survival threshold under $H_2O_2$ treatment than under glucose starvation. This at least partly explains

why moderate overexpression of SLC7A11 fails to induce cell death under $H_2O_2$ treatment but induces strong cell death under glucose starvation.

In the context of the current literature, which indicates that SLC7A11 has a protective role under oxidative stress conditions[8], our finding that SLC7A11 overexpression promotes $H_2O_2$-induced cell death is surprising and counterintuitive. We propose that SLC7A11 has opposing effects on cell survival and death under $H_2O_2$ treatment. SLC7A11-mediated cystine import increases intracellular cysteine and GSH reserves, which help cells to cope with $H_2O_2$-induced oxidative stress; however, the reduction of cystine to cysteine consumes

**Fig. 5 | High overexpression of SLC7A11 promotes primary tumor growth but suppresses tumor metastasis. a** Measurement of tumor volumes of indicated 786-O xenograft tumors after subcutaneous injection ($n = 8$ mice). **b** Schematic showing the method of intracardiac injection in mice. **c** Quantification of photon flux (photons per second) in mice normalized to day 0 after intracardiac injection of indicated 786-O cells ($n = 9$ mice). **d** Images of bioluminescence in mice 5 weeks after intracardiac injection of indicated 786-O cells (left) and statistical analysis of the whole-body photon flux (photons per second) (right) ($n = 9$ mice).
**e** Representative images of liver metastasis from SLC7A11-low, -moderate, and -high 786-O cells. **f** Representative images of hematoxylin and eosin (H&E) and immunohistochemical staining (SLC7A11) of livers with tumor metastasis derived from indicated 786-O cells. Scale bars, 20 μm. "T" stands for tumor cells. **g** Quantification of photon flux in mice normalized to day 0 after intracardiac injection of SLC7A11-low 786-O cells with PBS and -high 786-O cells treated with PBS, NAC, or Trolox

($n = 4$ mice for Low + PBS and High + Trolox groups, $n = 5$ mice for High + PBS and High + NAC groups). **h** Quantification of photon flux in mice normalized to day 0 after intracardiac injection of indicated H1299 cells ($n = 7$ mice). **i** Images of bioluminescence in mice 30 min after intracardiac injection of indicated H1299 cells (left) and statistical analysis of whole-body photon flux (right). ($n = 7$ mice). **j** Representative images of liver metastasis from SLC7A11-low, -moderate, and -high H1299 cells. **k** Analysis of *SLC7A11* expression levels between breast primary tumors and circulating tumor cells (CTCs) from matched patients with stage II-III breast cancer (GSE111842; $n = 12$ for primary tumors and $n = 16$ for CTCs). **l** Analysis of *SLC7A11* expression in estrogen receptor–positive (ER$^+$) breast primary tumors (PTs; from The Cancer Genome Atlas) and CTCs (GSE75367 and GSE86978; $n = 1015$ for PTs and $n = 99$ for CTCs). Data were presented as mean ± SD. *P* value was determined by two-tailed unpaired Student's *t* test. n.s. not significant. Source data are provided as a Source Data file.

NADPH, which can induce disulfide stress and cell death when coupled with the NADPH-depleting effect of $H_2O_2$ treatment (Fig. 6a). We further propose that SLC7A11's pro– or anti–cell death effect under $H_2O_2$ treatment is dictated by its expression level. Specifically, moderate overexpression of SLC7A11 in cancer cells appears to be beneficial, in that the antioxidant effect of GSH appears to be stronger than the NADPH-depleting effect of cystine reduction; consequently, moderate SLC7A11 overexpression suppresses $H_2O_2$-induced cell death (Fig. 6b). In contrast, in cancer cells with high SLC7A11 overexpression, drastic accumulation of intracellular cystine and other disulfide molecules under $H_2O_2$ treatment overrides any beneficial effect of GSH and leads to redox system collapse and rapid cell death (Fig. 6c).

Interestingly, increased cystine import has also been shown to promote $H_2O_2$-induced cell death in *Escherichia coli*; of note, SLC7A11 is not conserved in *E. coli*, which uses a different transporter, cysB, to mediate cystine import[32]. Mechanistically, it was proposed that, once intracellular cystine is reduced to cysteine, high levels of cysteine may promote the conversion of $H_2O_2$ to highly toxic hydroxyl radicals through the Fenton reaction, thereby promoting cell death[32]. However, we found that treatment with the iron chelator deferoxamine (DFO) failed to rescue $H_2O_2$-induced cell death in SLC7A11-high cancer cells used in our studies (Supplementary Fig. 6a); as a control, DFO treatment abolished RSL3-induced ferroptosis in these cell lines (Supplementary Fig. 6b). Therefore, high cystine import–induced cellular toxicity under $H_2O_2$ treatment is evolutionarily conserved, but cells apparently have developed different underlying mechanisms during evolution.

SLC7A11-mediated cystine uptake plays an important role in antioxidant defense, including ferroptosis mitigation, which is beneficial for tumor growth, and SLC7A11 is overexpressed in multiple human cancers[8]. Consistent with this, our data showed that high overexpression of SLC7A11 promoted primary tumor growth. However, our studies also revealed that high overexpression of SLC7A11 suppressed tumor metastasis, and CTCs exhibited lower SLC7A11 expression than did primary tumor samples from patients, which may seem counterintuitive in light of SLC7A11's established tumor-promoting effect. We propose that this may be due to the excessive susceptibility of SLC7A11-high cancer cells to cell death induced by oxidative stress during metastasis. Notably, while primary cancer cells generally exhibit high oxidative stress, metastasizing cancer cells often encounter even greater oxidative stress due to the hostile microenvironment they face as they break away from the primary tumor and travel to distant sites in the body[4,27]. Therefore, the susceptibility of SLC7A11-high cancer cells to oxidative stress may be more pronounced in metastasizing cancer cells than in primary tumor cells. This might help explain, at least partly, the differential effects of SLC7A11 high overexpression in primary tumor growth versus tumor metastasis. Our current data further suggest that SLC7A11-high cancer cells would be positively selected in primary tumors but negatively selected in metastasized tumors. Notably, completely ablating SLC7A11

expression also suppressed tumor metastasis[33], suggesting that moderate expression levels of SLC7A11 may be most beneficial for tumor metastasis, although further investigations are required to fully test this hypothesis. Further understanding this context-dependent function of SLC7A11 in cancer might provide important insights into therapeutic targeting of the disulfide stress–induced metabolic vulnerability in certain SLC7A11-high cancers.

## Methods
This research complies with all relevant ethical regulations of The University of Texas MD Anderson Cancer Center, including the Institutional Review Board and Institutional Animal Care and Use Committee.

### Cell culture studies
H1299 (CRL-5803), 786-O (CRL-1932), A498(HTB-44), H226 (CRL-5826), A549(CRL-7909), T98G(CRL-1690), Hs578T(HTB-126), and HEK293T (CRL-3216) cell lines were obtained from ATCC. UMRC6 (#08090513) was purchased from Sigma. All cell lines used in this study were free of mycoplasma contamination (per testing done by the vendor). All the cells were cultured in a 37 °C incubator in a 5% $CO_2$ atmosphere. H1299 cells were cultured in RPMI-1640 medium supplemented with 10% fetal bovine serum and 10,000 U/mL penicillin-streptomycin. 786-O, A498, UMRC6, H226, A549, T98G, Hs578T, and HEK293T cells were cultured in Dulbecco modified Eagle medium (DMEM) supplemented with 10% fetal bovine serum and 10,000 U/mL of penicillin-streptomycin. For the glucose deprivation experiments, cells were cultured in glucose-free DMEM (Life Technologies #11966025) supplemented with dialyzed fetal bovine serum as described previously[34,35].

### Reagents
XenoLight D-luciferin - K+ salt bioluminescent substrate (#122799) and [$^{14}C$] cystine (#NEC854010UC) were purchased from PerkinElmer. Erastin (#S7242), CB-839 (#S7655) and Z-VAD-FMK (#S7023) were purchased from Selleckchem. Necrostatin-1s (#2263-1) was purchased from BioVision. Staurosporine (#81590) was purchased from Cayman Chemical. The following reagents were obtained from Sigma-Aldrich: hydrogen peroxide (#216763-500 ML), TCEP (#C4706), 2-ME (#M6250), NAC (#A9165), D-penicillamine (#P4875), L-penicillamine (#196312), 2-deoxy-D-glucose (#D8375-1G), sulfasalazine (#S0883-10G), Trolox (#238813), DFO mesylate salt (#D9533), and ferrostatin-1 (#SML0583). All reagents were dissolved following the manufacturers' instructions.

### Constructs and generation of overexpression, knockdown, and knockout cell lines
To generate cell lines with stable overexpression of C-terminal–tagged SLC7A11, the *SLC7A11* cDNA was cloned into the lentivirus expression vector pLVX-C-Myc or -V5. HEK293T cells were transfected with either pLVX-empty vector, -SLC7A11-Myc, or -SLC7A11-V5, together with the

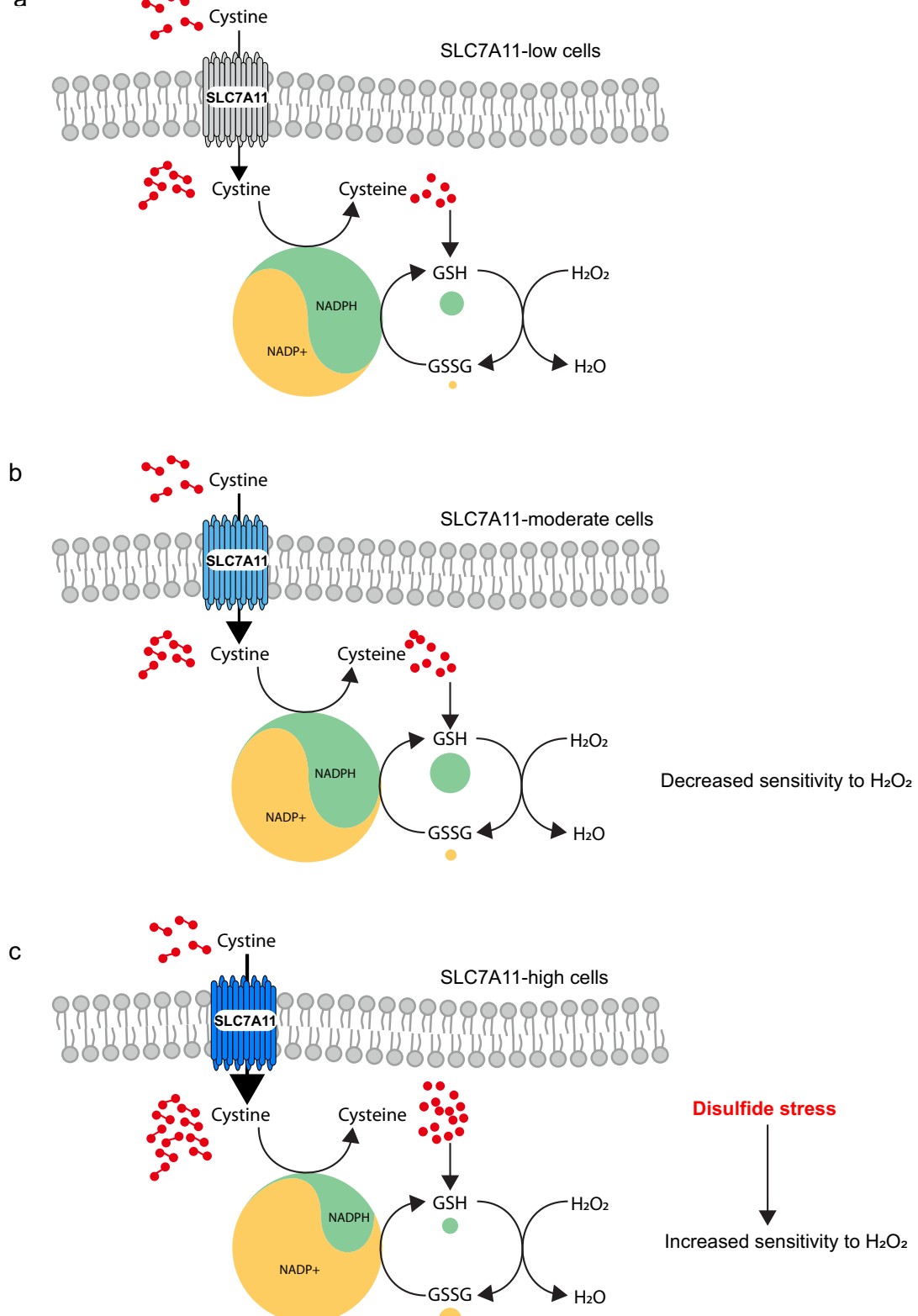

**Fig. 6 | Working model depicting how different expression levels of SLC7A11 dictate differential responses to oxidative stress in cancer cells. a** SLC7A11 imports cystine into cells to produce GSH, which can detoxify $H_2O_2$. However, both the reduction of cystine to cysteine and that of GSSG to GSH consume NADPH. **b** In cells with moderate expression of SLC7A11, the beneficial effect of GSH to detoxify $H_2O_2$ appears to be stronger than the NADPH-depleting effect of cystine reduction, resulting in decreased sensitivity to $H_2O_2$. **c** In cells with high expression of SLC7A11 and high cystine uptake, $H_2O_2$ treatment leads to drastic accumulation of intracellular cystine and other disulfide molecules and NADPH depletion, which overrides the beneficial effect of GSH and triggers rapid disulfidptosis. This explains the increased sensitivity to $H_2O_2$ in SLC7A11-high cells. NADPH nicotinamide adenine dinucleotide phosphate, GSH reduced glutathione, GSSG glutathione disulfide.

lentiviral packaging plasmids psPAX2 and pMD2.G using Lipofecta-mine 2000 reagent (Life Technologies, #11668030) according to the manufacturer's instructions and as described previously[36,37]. Forty-eight hours later, lentivirus was collected and filtered, and the target cell lines were infected with lentivirus. Twenty-four hours after infection, antibiotic was added to the medium to obtain stable cell lines. To generate *SLC7A11*-knockdown cell lines, lentiviral transduction with *SLC7A11* shRNA vectors was conducted as described previously[38]. Forty-eight hours later, upon puromycin antibiotic selection, SLC7A11 expression levels were determined by immunoblotting. To generate CRISPR knockout cells, guide RNAs were generated in LentiCRISPR-V2 (Addgene, #52961) according to the standard protocol, as described previously[39,40]. The knockout cells were identified by immunoblotting. The sequences of the single-guide RNAs are listed in Supplementary Table 1.

### Cell death assays
Cell death was measured using flow cytometry as described previously[41,42]. Briefly, cells were seeded in 12-well plates and incubated overnight. After treatment, cells were trypsinized and collected. After washing with cold phosphate-buffered saline (PBS), the cells were stained with 2 µg/mL propidium iodide (PI) in cold PBS. The fraction of dead cells (PI-positive cells) was measured using a BD Accuri C6 flow cytometer (BD Biosciences) or Attune NxT Flow Cytometer and analyzed by FlowJo 10 software[43,44]. All experiments were performed in triplicate.

### NADP$^+$ and NADPH measurement
The intracellular levels of NADPH and total NADP (NADPH + NADP$^+$) were measured as described in our previous publication[12]. Briefly, cells were seeded in 6-well plates and incubated overnight. After treatment, cells were lysed in 300 µL extraction buffer (20 mM nicotinamide, 20 mM NaHCO$_3$, 100 mM Na$_2$CO$_3$) and centrifuged, and the supernatant was split into two 150-µL aliquots. For the measurement of total NADP, 20 µL supernatant from one 150 µL aliquot and 80 µL of NADP-cycling buffer (100 mM Tris-HCl [pH8.0], 0.5 mM thiazolyl blue, 2 mM phenazine ethosulfate, 5 mM ethylenediaminetetraacetic acid [EDTA]) containing 1.0 U of G6PD enzyme (Sigma-Aldrich, #G4134) were mixed into a 96-well plate. After incubation for 1 min in the dark at 30 °C, 20 µL of 10 mM fresh glucose 6-phosphate solution was added to the mixture, and the change of absorbance at 570 nm was measured with a microplate reader every 30 s for 5 min at 30 °C. For the NADPH measurement, the remaining 150 µL supernatant was incubated at 60 °C for 30 min (to destroy NADP$^+$ without affecting NADPH), followed by the same procedures as those performed for the measurement of total NADP. Eventually, the concentration of NADP$^+$ was calculated by subtracting [NADPH] from [total NADP]. All experiments were performed in triplicate.

### ROS measurement
ROS measurement was performed as described previously[41]. Briefly, cells cultured in 12-well plates were incubated with media containing 2.5 µM of the ROS dye CM-H$_2$DCFDA (Life Technologies, #C6827) for 30 min at 37 °C. Then cells were trypsinized and collected. After washing with PBS, the cells were resuspended in PBS and subjected to flow cytometry analysis. All experiments were performed in triplicate.

### Cystine uptake assay
Cystine uptake was measured as described previously[45]. Briefly, cells were seeded in 12-well plates and incubated overnight. On the second day, the medium was replaced with fresh DMEM (which contains 5 µM cystine and [$^{14}$C] cystine [0.04 µCi]), and cells were incubated for the indicated time periods. Then the cystine uptake was terminated by rapidly rinsing cells with cold PBS and lysing them in 200 µL 0.1 mM NaOH solution. Radioactivity (DPM) was measured using a Tri-Carb

Liquid Scintillation Analyzer (PerkinElmer, Model 4810TR) in the presence of quench curve. All experiments were performed in triplicate.

### Western blotting
Western blotting was performed as described previously[46,47]. Briefly, cells were lysed in NP-40 lysis buffer (50 mM Tris [pH 7.4], 250 mM NaCl, 5 mM EDTA, 50 mM NaF, 1 mM Na$_3$VO$_4$, 1% Nonidet P40) containing complete mini protease inhibitors (Roche) and a Phosphatase Inhibitor Cocktail Set (Millipore). The protein concentration of lysates was quantified using a BCA Protein Assay Kit from Life Technologies (#23227). 30 µg of total protein was loaded into SDS-PAGE gel and transferred to a PVDF membrane (Bio-Rad) using standard techniques. The primary antibodies and concentrations used for Western blotting were Vinculin (1:2000, Sigma-Aldrich, #V4505), SLC7A11 (1:1000, CST, #12691S), SLC7A9 (1:1000, Thermo Fisher, PA5-50887), SLC3A1 (1:1000, Abcam, ab196552), GPX1 (1:1000, CST, 3286S), FLAG (1:1000, Sigma, F1804), BAX (1:1000, CST, #2772T), and BAK (1:1000, CST, #12105T).

### Intracellular GSSG and GSSG/GSH ratio detection in cells
The GSH/GSSG-Glo™ Assay kit (Promega, V6611) was used to measure intracellular GSSG levels and the GSSG/GSH ratio. Briefly, cells were seeded in a 96-well plate, followed by indicated treatment. Cells were then lysed either with Total or Oxidized Glutathione reagent and shaken for 5 mins. After the addition of the Luciferin Generation Reagent, cells were incubated for 30 mins. After the addition of luciferin, cells were incubated for another 15 mins. The luminescent signal was subsequently measured by a Gen5 microplate reader (Biotek).

### Intracellular glutamate detection in cells
The Glutamate-Glo™ Assay kit (Promega, J7021) was used to measure intracellular glutamate levels. Briefly, cells were seeded in a 96-well plate, followed by indicated treatment. Cells were then washed with ice-cold PBS and lysed with 0.6 N HCl. The lysing reagent was inactivated by adding 1 M Tris base, followed by the addition of the prepared Glutamate Detection Reagent and the incubation for 1 hr at room temperature. The luminescent signal was subsequently measured by a Gen5 microplate reader (Biotek).

### Cysteine detection in medium
The cysteine level was detected by using the Cysteine Assay Kit (Fluorometric) (Sigma, MAK255). Briefly, indicated regents (such as TCEP) were added into medium. Following a centrifuge at 10,000×$g$ for 5 mins, the supernatant was collected. Then 10 ul supernatant was plated for each sample in 96-well plate. Master Reaction Mix was added to each well followed by incubation for 30 mins. Then CYS probe was added to each well and fluorescence was measured by a Gen5 microplate reader (Biotek).

### Quantification of intracellular cystine, cysteine, glutathione, and glutathione disulfide levels using high-performance liquid chromatography-mass spectrometry
To accurately quantify the intracellular levels of various thiol species, we used an improved extraction and sample derivatization method to prevent thiols' oxidation before analysis, as recently described[48]. Cells cultured in 6-well plates were quickly rinsed with 1 mL of pre-cold PBS and then extracted in 500 µL of pre-chilled extraction buffer containing 40% acetonitrile, 40% methanol, and 20% water with the addition of 1 mM EDTA and 100 mM formic acid; the addition of EDTA prevents potential oxidation induced by metal ions, while formic acid prevents the formation of the highly reactive thiolate anion. Cells were incubated on ice for about 5 min and then scraped into pre-chilled 1.5 mL Eppendorf tubes and spun in a centrifuge for 10 min at 4 °C. The supernatant was collected into a new tube and stored with dry ice until

further analysis. The levels of cysteine and glutathione can be maintained for at least 48 h by this method.

For analysis, 90 μL of the supernatant was mixed with 10 μL of a mixture of isotopically labeled internal standards (U-$^{13}$C-$^{15}$N-cysteine, 1 μg/mL; U-$^{13}$C-$^{15}$N-cystine, 10 μg/mL; $2 \times ^{13}$C-1 $\times ^{15}$N-glutathione, 50 μg/mL; and $4 \times ^{13}$C-2 $\times ^{15}$N-glutathione disulfide, 10 μg/mL) in the same buffer. The mixture was then derivatized with benzyl chloroformate, which can form stable carboxybenzyl adducts with free amines and thiols. Next, we added 10 μL triethylamine and 1 μL benzyl chloroformate and incubated the mixture at 37 °C for 10 min. The derivatized samples were further analyzed using high-performance liquid chromatography–mass spectrometry, and concentrations were quantified according to standard curves. To determine the intracellular concentrations, cell volumes for each sample were determined using packed cell volume tubes through additional replicates.

### Cell line-derived xenograft experiments
The experiments with cell line–derived xenograft models were performed as described previously[49] and under the guidance of a protocol approved by the Institutional Animal Care and Use Committee and Institutional Review Board at The University of Texas MD Anderson Cancer Center. Mice were housed under specific-pathogen-free conditions with a 12 h light–12 h dark cycle. The ambient temperature was 21–23 °C, with 45% humidity and the mice had ad libitum access to water and food. For the 786-O cell line–xenograft experiments, 4- to 6-week-old female homozygous (*Foxn1$^{nu}$/Foxn1$^{nu}$*) nude mice were purchased from the Experimental Radiation Oncology Breeding Core Facility at MD Anderson and were kept in the Animal Care Facility at the Department of Veterinary Medicine and Surgery. *SLC7A11*-low, -moderate and -high 786-O cells were resuspended in fetal bovine serum–free DMEM, and the same numbers of cells were injected into mice subcutaneously after anaesthetization. The tumor volume was measured once per week until the endpoint and was calculated using the equation volume = length × width$^2$ × 1/2. The maximal tumor burden permitted by the ethics committee is a length of 1.5 cm, and the maximal tumor burden did not exceed the limit.

### Metastasis model
The experiments with metastasis models were performed under the guidance of a protocol approved by the Institutional Animal Care and Use Committee and Institutional Review Board at The University of Texas MD Anderson Cancer Center. For the metastasis experiments, 6- to 7-week-old female NSG mice were purchased from the Experimental Radiation Oncology Breeding Core Facility at MD Anderson and were kept in the Animal Care Facility at the Department of Veterinary Medicine and Surgery. In all, $1 \times 10^6$ cells (786-O) or $5 \times 10^5$ cells (H1299) were resuspended in 100 μL of PBS and injected into the left ventricle of the heart using a nonsurgical method and a 26G syringe. After 30 min, anesthetized mice were placed in the IVIS 200 Imaging System or IVIS Lumina and underwent whole-body imaging after intraperitoneal injection of the luciferase substrate D-luciferin. A successfully intracardiac injected mice was indicated by systemic bioluminescence distributed throughout the whole body. Only mice with satisfactory injections were used in the following experiment. Weekly bioimaging was used to assess the metastasis of the injected cells until the indicated time. For treatment with NAC or Trolox, all compounds were administered by i.p. injection every 48 h. NAC was dissolved in PBS and injected at a concentration of 120 mg/kg. Trolox was dissolved in PBS and injected at a concentration of 10 mg/kg.

### Bioluminescent imaging
For the in vivo metastasis experiments, cells were pre-labeled with luciferase fused with RFP. For the in vivo bioluminescent imaging, mice were injected with the luciferase substrate D-luciferin and anesthetized. Then mice were placed onto the ready-set IVIS 200 or IVIS Lumina Imaging System. Generally, five or three mice were imaged at a time. The signals emitted from the injected, luciferase-labeled cells were detected by the IVIS system and were analyzed and quantified as photon counts using Living Image software (Xenogen).

### Histopathological analysis
Mouse tissue and xenograft tumor samples were collected and immediately fixed in 10% neutral-buffered formalin overnight. After they were washed once with PBS, the samples were transferred into 70% ethanol and were embedded, sectioned, and stained with hematoxylin and eosin by the Center for Radiation Oncology Research at MD Anderson. For the immunohistochemistry staining, sample sections were processed according to the methods described in our previous publications[50–52]. The primary antibodies used for immunohistochemical analysis were anti-SLC7A11 (1:800, Novus, NB300-318), anti-Ki-67 (D2H10; 1:500, CST, 9027 S), and anti-cleaved caspase-3 (1:500, CST, 9661S). Images were obtained at 400 × magnification using an Olympus microscope (BX43). All the histopathological analyses were conducted by Dr. James You, a board-certified pathologist.

### Analysis of *SLC7A11* expression in tumors
Data on the expression of *SLC7A11* in ER$^+$ breast primary tumors were downloaded from The Cancer Genome Atlas database. GSE111842 was downloaded from the Gene Expression Omnibus database to analyze SLC7A11 expression. Data on the expression of SLC7A11 in CTCs were downloaded from ctcRbase (http://www.origin-gene.cn/database/ctcRbase/).

### Statistics and reproducibility
The results of the cell culture experiments were collected and analyzed from at least three independent repeats. Volumes or photon counts from at least four tumors or mice in each group were plotted. For immunoblots, the experiments have been repeated at least twice with similar results and representative data was shown. Data are presented as means ± standard deviations (s.d.). Statistical significance (*P* value) was calculated using a two-tailed, unpaired Student's *t* test by GraphPad Prism 9.0. A *P* value < 0.05 was considered significant. No statistical method was used to pre-determine the sample size. No data were excluded from the analyses. Unless stated otherwise, the experiments were not randomized and the investigators were not blinded to treatment allocation during experiments or to the outcome assessment.

### Reporting summary
Further information on research design is available in the Nature Portfolio Reporting Summary linked to this article.

## Data availability
The uncropped films for immunoblots used in this study are shown in Source Data files. All data were available within the paper and Source Data file. Source data are provided with this paper.

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

## Acknowledgements

We thank Laura L. Russell from the Research Medical Library at The University of Texas MD Anderson Cancer Center for editing the manuscript. This research was supported by the Institutional Research Fund and Bridge Fund from MD Anderson Cancer Center; the Emerson Collective Cancer Research Fund; the Cancer Prevention & Research Institute of Texas grants RP220258 and RP230072; and R01CA181196, R01CA244144, R01CA247992, R01CA269646, and U54 CA274220 from the National Institutes of Health; and the N.G. and Helen T. Hawkins Distinguished Professorship for Cancer Research of The University of Texas MD Anderson Cancer Center (to B.G.); US National Institutes of Health grants R01CA166051 and R01CA269140; an American Cancer Society grant (award number: DBG-22-161-01-MM); and the Nylene Eckles Distinguished Professorship of The University of Texas MD Anderson Cancer Center (to L.M.). This research was also supported by the National Institutes of Health Cancer Center Support Grant (P30CA016672) to MD Anderson.

## Author contributions

Y.Y. performed most of the experiments with assistance from G.L., A.H., X.L., C.M., S.W., and L.Z.; K.O. and L.K. conducted metabolomics analyses; H.T. and Q.H. provided technical support for the metastasis experiments under the guidance of L.M.; M.V.P. provided resources for the metabolomics analyses; M.J.Y. performed the histopathological analysis. B.G. and Y.Y. designed the experiments; B.G. supervised the project, provided funding support, and established collaboration; B.G. and Y.Y. wrote the manuscript; all authors commented on the manuscript.

## Competing interests

K.O. and L.K. are former full-time employees of Kadmon Corporation and are now full-time employees of the Barer Institute and Sanofi, US, respectively. M.V.P. is a full-time employee of Kadmon Corporation, a Sanofi Company. The other authors declare no competing interests.
