## [Peer Review File · Nature Communications]

REVIEWER COMMENTS

Reviewer #1 (Remarks to the Author):

The work by Yan et al describes a liability for cells that highly express SLC7a11, the transporter of cystine, the oxidized, disulfide conjugated form of two cysteines. The author demonstrated that high SLC7a11 expression sensitizes cells to NADPH depletion due to high cystine uptake. Intracellularly, cystine is being reduced to two cysteines by reduced glutathione (GSH) which itself enter a redox cycle by first generating the disulfide oxidized glutathione, GSSG, and then, in an NADPH-dependent manner, is being reduced back to GSH. Hence, the higher demand to NADPH in SLC7a11 high cells, and the sensitivity to NADPH depletion.

It should be noted that the above-described mechanism is not a novel discovery of the authors. Furthermore, the authors showed in the past that cells expressing high levels of SLC7a11 are susceptible to glucose deprivation that causes a decrease in NADPH production in the pentose phosphate pathway (PPP). So, one needs to consider whether the new results described in this manuscript are sufficiently novel to warrant publication in Nature Communications or whether they are incremental steps in our scientific knowledge.

The novel discoveries of this work can be summarized as follow:

1. Uncontrolled high rate of cystine uptake and reduction to cysteine due to ultra-high levels of SLC7a11 sensitizes cells to NADPH depletion by induced redox stress.
2. Death induced by NADPH depletion is not directly mediated by ROS, but rather by the intracellular accumulation of disulfide molecules such as cystine, GSSG and others.
3. While high levels of SLC7a11 provide a growth advantage to cells and tumors, they inhibit metastases, likely due to the redox stress associated with the metastatic process.

Specific comments:

1. The levels of SLC7a11 is dramatically diverse between the different cells studied, hence, it is not possible to make general definitions of 'low', 'moderate' or 'high' expression of SLC7a11. In H1299, the so-called 'moderate' levels of the protein seem massively higher than any other 'high' levels in other cell lines. There is also a lack of clear correlations between cell lines between the levels of SLC7a11 and cystine uptake. This either question the main point of this work, or suggest that other unidentified mechanisms link SLC7a11 to hydrogen peroxide sensitivity.

2. The role of GSH-dependent peroxidases should be investigated in the process of depleting GSH and increasing disulfide molecules in hydrogen peroxide treated SLC7a11 'high' cells.

3. While SLC7a11 is important for cystine uptake, it excrete glutamate at the same time. The effect of high levels of SLC7a11 on glutamate metabolism was ignored in this work. Glutamate, like cysteine, is required for glutathione biosynthesis, and to many other intracellular functions beyond protein synthesis.

4. The direct consequences of xCT inhibition with erastin, or the reduction of extracellular cystine with reducing agents such as 2ME, was not tested on extracellular levels of cystine. This should be done to confirm the on-target effectiveness of the treatment.

5. The use of TPNOX to deplete NADPH without hydrogen peroxide is an elegant way to strengthen the authors' claim to the manner by which uncontrolled cystine uptake sensitizes cells to redox stress. A question remained is whether TPNOX would prevent metastasis of SLC7a11 'medium' cells?

Reviewer #2 (Remarks to the Author):

In this manuscript "SLC7A11 expression level dictates differential responses to oxidative stress in cancer cells", Yuelong Yan and his colleagues found moderate overexpression of the cystine

transporter solute carrier family 7 member 11 (SLC7A11) is beneficial for cancer cells treated with H₂O₂, its high overexpression increases H₂O₂-induced cell death due to accumulation of intracellular cystine and other disulfide molecules like GSSG, NADPH depletion, redox system collapse, and cell death. Moreover, high overexpression of SLC7A11 promotes tumor growth but suppresses tumor metastasis.

There are a few concerns/questions listed below:

1. The human tumor cells used in this manuscript are low expression of endogenous SLC7A11 cells - H1299 (non-small-cell lung cancer cell), 786-O (renal clear cell carcinoma cell), high expression of endogenous SLC7A11 cells - T98-G (Glioma cell), Hs578T (breast cancer cell). To comprehensively determine whether SLC7A11 plays a key role in cystine transport, the expression and function of other cysteine transporters like SLC3A1 should be in consideration with detection and relative assays, it is necessary to exclude the heterogeneity of tumor and related mechanisms as well as the specificity of transporters.
2. How to define "low/moderate/high overexpression of SLC7A11"? It is suggested that all relevant WB bands in the paper be quantified and compared with cysteine uptake level, GSH/GSSG ratio and other key data to find correlations (fig1a-f, s1a-1b).
3. It is necessary to consider whether gene overexpression and other cell construction methods may lose some expression during long-term cell culture, which will cause instability of the results.
4. The influence of moderate overexpression of SLC7A11 either on cell death (fig1), intracellular cystine concentration (fig1a), or NADP⁺/NADPH ratio (fig4a), or on the growth and metastasis of the mouse tumor model (fig5a-d) have inconsistent changes in the high overexpression of SLC7A11, the results seem strange, the authors interpret in the discussion section said that, "We further propose that SLC7A11's pro- or anti-cell death effect under H₂O₂ treatment is dictated by its expression level. Specifically, moderate overexpression of SLC7A11 in cancer cells appears to be beneficial, in that the antioxidant effect of GSH appears to be stronger than the NADPH-depleting effect of cystine reduction; consequently, moderate SLC7A11 overexpression suppresses H₂O₂-induced cell death (Fig. 6b)." But in fig2e, GSSG/GSH ratio seems have no change.
5. In fig1a-1d, low/moderate/high overexpression of SLC7A11 reduced cell death after administration of H₂O₂, high overexpression of SLC7A11 increased cell death, both moderate/high overexpression SLC7A11 increased cell death for glucose starvation. Cells with high endogenous expression of SLC7A11: knockout SLC7A11 increased cell death; knockdown SLC7A11 reduces cell death; for glucose starvation, both moderate/high overexpression SLC7A11 reduced cell death. But cysteine levels are not the cause of cell death; moreover the cause of cell death was not a normal one (like apoptosis, ferroptosis, etc.), and also not due to ROS (fig3&s3); what are the specific causes of cell death? And the relationship with glucose starvation and H₂O₂?
6. In fig2d and 2i, the concentration of GSSG in 786-O cells with high overexpression of SLC7A11 after H₂O₂ treatment was much lower than that in T98G cells (with endogenous high expression of SLC7A11) after H₂O₂ treatment (1mM VS 6mM). Does it indicate that there are other factors besides SLC7A11 that play a key role in the concentration of disulfide metabolites such as GSSG? Considering with follow-up questions, what concentration of GSSG, GSH and other metabolites is required to match the function and effect of endogenous high expression of SLC7A11? Similar problems appear in fig3a-e, the concentration of cystine and GSSG is also ten times higher than that in fig2a and 2d. In the same group, how to explain the inconsistency of the quantitative concentration of metabolites such as cystine, GSSG?
7. From the results of fig4g-j, not enough evidence could support the conclusion of "Together, our data suggest that high expression of SLC7A11 in combination with H₂O₂ treatment depletes NADPH, which contributes to H₂O₂-induced cell death."
8. As it mentioned in the paper "In 786-O xenograft tumor models, SLC7A11-high tumors exhibited significantly increased tumor growth compared with SLC7A11-moderate or -low tumors (Fig. 5a), which was consistent with our and others' previous findings showing that SLC7A11 promotes tumor growth by suppressing ferroptosis", the results of figs1e-f suggest that the use of ferroptosis inhibitors in 786-O and T98G has no effect on cell death in vitro, whether SLC7A11 overexpression or knockout, or with treatment of H₂O₂.
9. In this paper, only one breast cancer cell line were used in vitro, and tissue sample data of the Cancer Genome Atlas dataset and breast CTCs with unmatched control breast cancer tissue sample data were used to compare the expression level of SLC7A11. It seems not enough to say "Together, our data indicate that high SLC7A11 overexpression promotes primary tumor growth but suppresses metastasis, likely because SLC7A11-high cancer cells are susceptible to cell

death induced by oxidative stress during metastasis. "

Reviewer #3 (Remarks to the Author):

The paper is a follow up of a previous publication from this group examining the role of SLC7A11 in controlling cell death and survival. SLC7A11 imports cystine and has been shown to protect cells from excessive ROS and ferroptosis. However, reduction of cystine to cysteine requires NADPH and - as shown previously by this group - limiting glucose to high SLC7A11 expressing cells results in decreased NADPH production through the PPP and cell death due to disulfide accumulation. In the present paper the authors show that the level of SLC7A11 can determine the response to oxidative stress - with moderate expression protecting cells while high levels leading to cell death due to the accumulation of intracellular cystine.

The results presented largely follow a logical progression from the author's previous work. However, the physiological relevance of the work isn't quite clear.

1. The authors show that modulating the expression of SCL7A11 can result in changes of cystine uptake and increase or decrease of cell death. While the cell death associated with high SLC7A11 expression is shown not to be apoptosis or ferroptosis, it's not clear what the moderate expression of SLC7A11 is protecting from. Is this ferroptosis?
2. It seems that several cancer cell lines express high levels of SLC7A11 that support more cell death under both ROS and glucose starved conditions than seen in cells with moderately reduced SCL7A11. Can the authors speculate on why these tumour cells would be selected to carry such high expression - which seems to be generally detrimental to survival?
3. In 786-O cells, moderate and high expression of SLC7A11 leads to a similar increase in cystine uptake (Fig 1e). However, there is a dramatic difference in the intracellular levels of cysteine and survival of these cells in response to H₂O₂ (Figure 2a). Does cystine uptake also show this strong increase in H₂O₂ treated cells? What is regulating this increase in cystine uptake in response to H₂O₂?
4. Can the authors rescue cell death by replenishing NADPH in these cells? Maybe activation of another NADPH generating system?
5. High SLC7A11 expressing cells form much larger tumours in mice than the low or moderately expressing lines (Figure 5a). The authors suggest this is due to a protection from ferroptosis - I think it would be necessary to show that these cells are resistant to ferroptosis. What is inducing ferroptosis in these primary tumors, if not ROS or glucose depletion?
6. Given the dramatic difference in tumour growth it would be prudent to show the reproducibility of this with a different high SLC7A11 expressing line (H1299). Despite being significantly protected from ferroptosis, these cells are deficient in liver colonisation. Can this be rescued - for example with NAC but not with Trolox?

RESPONSE TO REVIEWERS' COMMENTS

Note to reviewers: We thank reviewers for taking efforts to review our manuscript and for providing insightful comments to further improve our manuscript. Below we provide the detailed point-by-point response to address all the comments raised by reviewers. To facilitate the review of our rebuttal letter and manuscript by reviewers, we present all the new data as rebuttal letter figures in this letter, with referrals to corresponding figures and text in our revised manuscript. We have also marked all the changes in our revised manuscript by colored text.

REVIEWER COMMENTS

Reviewer #1 (Remarks to the Author):

The work by Yan et al describes a liability for cells that highly express SLC7a11, the transporter of cystine, the oxidized, disulfide conjugated form of two cysteines. The author demonstrated that high SLC7a11 expression sensitizes cells to NADPH depletion due to high cystine uptake. Intracellularly, cystine is being reduced to two cysteines by reduced glutathione (GSH) which itself enter a redox cycle by first generating the disulfide oxidized glutathione, GSSG, and then, in an NADPH-dependent manner, is being reduced back to GSH. Hence, the higher demand to NADPH in SLC7a11 high cells, and the sensitivity to NADPH depletion.

It should be noted that the above-described mechanism is not a novel discovery of the authors. Furthermore, the authors showed in the past that cells expressing high levels of SLC7a11 are susceptible to glucose deprivation that causes a decrease in NADPH production in the pentose phosphate pathway (PPP). So, one needs to consider whether the new results described in this manuscript are sufficiently novel to warrant publication in Nature Communications or whether they are incremental steps in our scientific knowledge.

The novel discoveries of this work can be summarized as follow:

- 1. Uncontrolled high rate of cystine uptake and reduction to cysteine due to ultra-high levels of SLC7a11 sensitizes cells to NADPH depletion by induced redox stress.*
- 2. Death induced by NADPH depletion is not directly mediated by ROS, but rather by the intracellular accumulation of disulfide molecules such as cystine, GSSG and others.*
- 3. While high levels of SLC7a11 provide a growth advantage to cells and tumors, they inhibit metastases, likely due to the redox stress associated with the metastatic process.*

We thank the reviewer for summarizing and commenting on our manuscript, and kindly ask the reviewer to consider the following additional points while evaluating the conceptual novelty of our study.

Based on the current dogma, SLC7A11 has a well-established protective role under oxidative stress conditions (including H₂O₂-induced oxidative stress). Therefore, our finding that high expression of SLC7A11 dramatically promotes H₂O₂-induced cell death is counterintuitive and novel, representing a paradigm shift from the current understanding of SLC7A11 in redox biology. Importantly, we provide a mechanistic explanation on why different expression levels of SLC7A11 have opposing effects on cell survival and death under H₂O₂ treatment, which will

inspire additional research into understanding SLC7A11-mediated redox maintenance. In addition, our finding that high expression of SLC7A11 suppresses metastasis is also surprising.

Of note, while high SLC7A11-induced cell death under glucose starvation (1) vs H₂O₂ treatment (from this study) shares some common underlying mechanisms (such as NADPH depletion and aberrant accumulation of disulfide molecules), our previous data *per se* (1) do not predict that high SLC7A11 would have a potent cell death-promoting effect under H₂O₂ treatment. Therefore, we argue that our current findings are not merely incremental, and hope the reviewer agrees that these conceptual novelties justify its publication at the level of *Nature Communications*.

Specific comments:

1. The levels of SLC7a11 is dramatically diverse between the different cells studied, hence, it is not possible to make general definitions of 'low', 'moderate' or 'high' expression of SLC7a11. In H1299, the so-called 'moderate' levels of the protein seem massively higher than any other 'high' levels in other cell lines. There is also a lack of clear correlations between cell lines between the levels of SLC7a11 and cystine uptake. This either question the main point of this work, or suggest that other unidentified mechanisms link SLC7a11 to hydrogen peroxide sensitivity.

In our study, the general definition of low-, moderate-, or high-expression of SLC7A11 is guided by relative expression levels of endogenous SLC7A11 and corresponding cystine uptake rate in diverse cancer cell lines. Taking the kind suggestion from this reviewer, we analyzed SLC7A11 protein levels and cystine uptake rates across a panel of cancer cell lines. As shown in **rebuttal letter Fig. 1A, B** (Fig. 1a, b in the manuscript), cystine uptake levels in general correlated with expression levels of SLC7A11 (but not with those of other proteins involved in cystine uptake, such as SLC7A9 and SLC3A1) in these cell lines. We further categorized these cell lines into SLC7A11-low, -moderate, and -high cells based on their relative SLC7A11 expression and cystine uptake levels, in which SLC7A11-moderate (UMRC6, H226, A498, and A549) and -high cell lines (T98G and Hs578T) exhibited 2-5- and 10-fold increases, respectively, in cystine uptake compared to SLC7A11-low cell lines (H1299 and 786-O) (such that SLC7A11-low cells exhibited cystine uptake levels at $\leq 10 \times 10^3$ DPM, SLC7A11-high cells at $\geq 55 \times 10^3$ DPM, whereas SLC7A11-moderate cells had cystine uptake levels between 10×10^3 and 55×10^3 DPM).

Importantly, knocking-down of SLC7A11 in SLC7A11-moderate or -high cells resulted in different phenotypes in terms of H₂O₂-induced cell death: while knocking down SLC7A11 in SLC7A11-moderate cells increased H₂O₂-induced cell death, as expected (**rebuttal letter Fig. 1C-F**; Fig. S1b, c in the manuscript), knocking down SLC7A11 in SLC7A11-high cells actually suppressed H₂O₂-induced cell death (see Fig. 1c-j in the manuscript). This is consistent with our model that high and moderate expression levels of SLC7A11 have opposing effect on H₂O₂-induced cell death (see Discussion at pages 21-22 in the manuscript).

Conversely, we can also overexpress SLC7A11 in SLC7A11-low cells (H1299 and 786-O) at different levels to achieve moderate or high cystine uptake levels, resulting in suppression or promotion of H₂O₂-induced cell death (see Fig. 1k-n in the manuscript). Here, it is important

Figure 1. Protein levels of SLC7A11, SLC7A9 and SLC3A1 (A) and corresponding cystine uptake levels (B) in a panel of cancer cell lines. (C-F) SLC7A11 knock down sensitizes SLC7A11-moderate cell lines to H₂O₂-induced cell death. **: P<0.01; *: P<0.001.**

to note that we had to overexpress SLC7A11 in SLC7A11-low cells to a higher level than that of endogenous SLC7A11 in SLC7A11-moderate cell lines in order to achieve a corresponding moderate increase of cystine uptake. (This is also an argument from the reviewer in the question above.) This is because SLC7A11 requires its partnering with the chaperone protein SLC3A2 in order to localize on the plasma membrane and mediate cystine uptake, and SLC3A2 also binds to multiple other amino acid transporters in cells; consequently, we need to overexpress SLC7A11 at higher levels to compete with other transporters for partnering with certain amount of SLC3A2 (in other words, some of the overexpressed SLC7A11 are nonfunctional because they do not bind to SLC3A2). Therefore, it can be misleading to cross-compare SLC7A11 levels between cell lines with SLC7A11 overexpression with other cell lines without SLC7A11 overexpression; instead, in this case, comparing their cystine uptake levels would be more accurate. We hope this reviewer agrees with our interpretation.

To summarize, (1) in cell lines without any genetic manipulation, definition of low-, moderate-, or high-expression of SLC7A11 can be guided by relative expression levels of

endogenous SLC7A11 and corresponding cystine uptake rate in these cell lines; and (2) for the reason listed above, it could be misleading to cross-compare SLC7A11 protein levels in cell lines with its overexpression vs those without overexpression. In this case, categorizing these cell lines based on their cystine uptake levels provide a more accurate and quantitative way to define these cell lines.

2. The role of GSH-dependent peroxidases should be investigated in the process of depleting GSH and increasing disulfide molecules in hydrogen peroxide treated SLC7a11 'high' cells.

The reviewer raised a very interesting question. Based on our model (**rebuttal letter Fig. 2A**; also see Discussion in the manuscript), we propose that, in H₂O₂-treated SLC7A11-high cells, NADPH depletion and disulfide stress-induced cell death are triggered by two NADPH-consuming processes, namely cystine reduction to cysteine and H₂O₂ detoxification by GSH-dependent peroxidases. According to this model, deleting GSH-dependent peroxidases (such as

Figure 2. GPX1 deletion suppresses H₂O₂-induced cell death in SLC7A11-high cells. (A) Simplified schematic illustrating redox systems regulated by SLC7A11-mediated cystine uptake. (B-E) GPX1 knock out alleviated H₂O₂-induced cell death (C), NADPH depletion (D), GSH depletion (E), increased GSSG level (F) and increased GSSG/GSH ratio (G) in SLC7A11-high cells. *: P<0.05; **: P<0.01; ****: P<0.0001.

GPX1) should alleviate NADPH depletion and cell death in H₂O₂-treated SLC7A11 high cells.

To test this hypothesis, we studied the role of GPX1 in H₂O₂-induced cell death in SLC7A11-high cells. As shown in **rebuttal letter Fig. 2** (Fig. 4f-j in the revised manuscript), we found that GPX1 deletion in SLC7A11-low 786-O cells increased H₂O₂-induced cell death (which is consistent with the protective role of GPX1 in detoxifying H₂O₂-induced oxidative stress) (**Panels B, C**); interestingly, GPX1 deletion in SLC7A11-high counterparts had an opposite effect and moderately suppressed H₂O₂-induced cell death (**Panels B, C**). Consistently, GPX1 deletion also alleviated NADPH depletion and GSSG level (as well as the GSSG/GSH ratio) increases in H₂O₂-treated SLC7A11-high cells (**Panel D-F**). It should be noted that the cell death reduction caused by GPX1 deletion in H₂O₂-treated SLC7A11-high cells was moderate (**Panel C**). Our interpretation is that, while GPX1 deletion reserves more NADPH and therefore protects SLC7A11-high cells from disulfide stress-induced cell death, reduction in H₂O₂ detoxification would still render these cells susceptible to H₂O₂-induced oxidative stress, eventually resulting in cell death.

Therefore, our data suggest that, similar to SLC7A11, GPX1 also has a dual role in regulating cell survival/death under oxidative stress. On one hand, GPX1 uses GSH as its co-factor to detoxify H₂O₂ and protects cells from H₂O₂-induced oxidative stress and cell death in SLC7A11-low cells. On the other hand, GPX1-mediated H₂O₂ detoxification consumes NADPH. When combined with high rates of cystine reduction in SLC7A11-high cells, this can lead to NADPH depletion, disulfide stress and subsequent cell death. Together, these data provide additional support for our model.

3. While SLC7a11 is important for cystine uptake, it excrete glutamate at the same time. The effect of high levels of SLC7a11 on glutamate metabolism was ignored in this work. Glutamate, like cysteine, is required for glutathione biosynthesis, and to many other intracellular functions beyond protein synthesis.

This reviewer raised an excellent point. Specifically, since SLC7A11 is an antiporter that imports cystine and exports glutamate, it can be argued that the decreased intracellular glutamate levels (caused by SLC7A11-mediated glutamate export) might also contribute to H₂O₂-induced cell death in SLC7A11-high cancer cells. Intracellular glutamate is mainly generated from glutamine through glutaminase (GLS; see **rebuttal letter Fig. 3A**). Therefore, this argument would predict that (1) GLS inhibition, similar to high expression of SLC7A11, should promote H₂O₂-induced cell death (by decreasing intracellular glutamate levels), and (2) further depleting intracellular glutamate levels by combining GLS inhibition with SLC7A11 high expression should exacerbate H₂O₂-induced cell death.

However, our following data did not support this premise (**rebuttal letter Fig. 3**; Fig. S4 a-d in the revised manuscript). Specifically, we found that, in SLC7A11-low cells, treatment with the GLS inhibitor CB-839 caused more decreases in intracellular glutamate levels than did high SLC7A11 overexpression (**Panel B**), but did not promote H₂O₂-induced cell death (**Panel C**).

Furthermore, while CB-839 treatment further decreased intracellular glutamate levels in SLC7A11-high cells (**Panel B**), CB-839 even attenuated H₂O₂-induced cell death in SLC7A11-high cells (**Panel C**). This last observation is consistent with our model that the cell death in H₂O₂-treated SLC7A11-high cells is caused by high cystine uptake in these cells, because decreasing intracellular glutamate levels by CB-839 treatment is expected to suppress SLC7A11-mediated cystine uptake.

Indeed, we confirmed the decrease of cystine uptake in CB-839-treated SLC7A11-high cells (**Panel D**). Together, these data support the model that the increased cell death in H₂O₂-treated SLC7A11-high cells is caused by cystine import, but not by glutamate export; consequently, suppressing cystine uptake by decreasing intracellular glutamate levels attenuates this cell death.

4. The direct consequences of xCT inhibition with erastin, or the reduction of extracellular cystine with reducing agents such as 2ME, was not tested on extracellular levels of cystine. This should be done to confirm the on-target effectiveness of the treatment.

We confirmed that erastin treatment potently suppressed cystine uptake (**rebuttal letter Fig. 4A**; Fig. S3e in the revised manuscript). The second question from the reviewer refers to the following statement in our manuscript “To this end, we took several approaches to preventing disulfide accumulation. These included ... (ii) treatment with the disulfide-reducing agents tris(2-carboxyethyl) phosphine (TCEP) and 2-mercaptoethanol (2-ME) to reduce cystine to cysteine in the medium (and thereby bypass SLC7A11-mediated cystine transport) (Fig. S3c).”

To address this question, we measured extracellular cysteine levels of SLC7A11-high cells treated with TCEP or 2-ME. Interestingly, our results showed that while TCEP treatment dramatically increased extracellular cysteine levels, as expected, 2-ME treatment only moderately increased extracellular cysteine levels (**rebuttal letter Fig. 4B**; Fig. S3f in the revised manuscript). This is because the mechanistic bases for how TCEP and 2-ME reduce extracellular cystine are somewhat different: TCEP directly cleaves cystine to two molecules of cysteine, whereas 2-ME reacts with cystine to generate a molecule of cysteine and a mixed

Figure 3. Glutaminase inhibition decreases intracellular glutamate levels and suppresses H₂O₂-induced cell death in SLC7A11-high cells. (A) Simplified schematic illustrating how glutamine metabolism is involved in SLC7A11-mediated cystine uptake. **(B-D)** CB-839 treatment decreases intracellular glutamate level **(B)**, suppresses H₂O₂-induced cell death **(C)**, and inhibits cystine uptake in SLC7A11-high cells **(D)**. *: P<0.05; ****: P<0.0001.

disulfide of 2-ME and cysteine (i.e., Cys-Cys + 2-ME → Cys + 2-ME-Cys). Because extracellular cysteine is unstable, it will continue to react with another molecule of 2-ME (or a 2-ME-2-ME disulfide), ending up with most products as 2-ME-Cys disulfide (2) (which cannot be detected in this assay).

Of note, extracellular 2-ME-Cys can be taken up into cells via system L (3) and is subsequently converted back to cysteine inside cells, thereby still bypassing SLC7A11-mediated cystine transport to provide intracellular cysteine for GSH synthesis. Since this mechanism is well established and 2-ME is widely used as a reducing agent in culturing medium to bypass SLC7A11-mediated cystine transport (for example, some SLC7A11 KO cell lines, such as mouse embryonic fibroblasts [MEFs], fail to survive in normal cell culture medium due to cystine uptake deficiency and ferroptosis induction, and 2-ME supplementation in culture medium can support the survival of SLC7A11 KO MEFs (4)), we hope this reviewer agrees that our interpretation and data are acceptable for our manuscript revision. To be more clarified, we have modified Fig. S3c to include 2-ME-Cys in the schematic (**rebuttal letter Fig. 4C**; Fig. S3c in the manuscript).

Figure 4. (A) Erastin treatment decreases cystine uptake in SLC7A11-high cells. (B) TCEP and 2-ME treatment increases cysteine levels in the media. (C) Schematic showing how 2-ME reacts with cystine to produce 2-ME-Cys. **: P<0.01; ****: P<0.0001.

5. The use of TPNOX to deplete NADPH without hydrogen peroxide is an elegant way to strengthen the authors' claim to the manner by which uncontrolled cystine uptake sensitizes cells to redox stress. A question remained is whether TPNOX would prevent metastasis of SLC7a11 'medium' cells?

We thank the reviewer for commenting that our TPNOX experiment is elegant and for asking an insightful question. This question refers to our data from Fig. 4k-m (also see **rebuttal letter Fig. 5A-B**), which shows that TPNOX overexpression increased the NADP⁺/NADPH ratio and

Figure 5. The effect of TPNOX overexpression in SLC7A11-low cancer cells and tumors. (A) Measurement of the NADP⁺/NADPH ratio in SLC7A11-low and -high 786-O cells with empty vector (EV) or TPNOX overexpression under vehicle or 1 mM H₂O₂ treatment. (B) Cell death measurement in response to treatment with 1 mM H₂O₂ for 6 hours in SLC7A11-low and -high 786-O cells with EV or TPNOX overexpression. (C) Quantification of photon flux (photons per second) in mice normalized to day 0 after intracardiac injection of SLC7A11-high 786-O cells with EV or TPNOX overexpression (n = 5 mice for each group). *: P<0.05; ****: P<0.0001.

promoted H₂O₂-induced cell death in SLC7A11-low cells. Since we overexpressed TPNOX in SLC7A11-low cells, we think it would make more sense to test the effect of TPNOX overexpression on metastasis of SLC7A11-low cancer cells (rather than SLC7A11-moderate cells). Our data showed that TPNOX overexpression moderately decreased metastasis in SLC7A11-low tumors (**rebuttal letter Fig. 5C**; Fig. S5e in the manuscript), which is consistent with the relatively moderate effect of TPNOX overexpression on depleting NADPH and promoting H₂O₂-induced cell death than did high overexpression of SLC7A11 (see **rebuttal letter Fig. 5A-B**).

Reviewer #2 (Remarks to the Author):

In this manuscript “SLC7A11 expression level dictates differential responses to oxidative stress in cancer cells”, Yuelong Yan and his colleagues found moderate overexpression of the cystine transporter solute carrier family 7 member 11 (SLC7A11) is beneficial for cancer cells treated with H₂O₂, its high overexpression increases H₂O₂-induced cell death due to accumulation of intracellular cystine and other disulfide molecules like GSSG, NADPH depletion, redox system collapse, and cell death. Moreover, high overexpression of SLC7A11 promotes tumor growth but suppresses tumor metastasis.

There are a few concerns/questions listed below:

1. The human tumor cells used in this manuscript are low expression of endogenous SLC7A11 cells - H1299 (non-small-cell lung cancer cell), 786-O (renal clear cell carcinoma cell), high expression of endogenous SLC7A11 cells - T98-G (Glioma cell), Hs578T (breast cancer cell). To comprehensively determine whether SLC7A11 plays a key role in cystine transport, the expression and function of other cysteine transporters like SLC3A1 should be in consideration with detection and relative assays, it is necessary to exclude the heterogeneity of tumor and related mechanisms as well as the specificity of transporters.

2. How to define “low/moderate/high overexpression of SLC7A11”? It is suggested that all relevant WB bands in the paper be quantified and compared with cysteine uptake level, GSH/GSSG ratio and other key data to find correlations (fig1a-f, s1a-1b).

Since these two questions are related to each other, we address them together below. To address these important questions raised by the reviewer, we measured cystine uptake levels and endogenous protein levels of SLC7A11 (as well as other proteins involved in cystine uptake, such as SLC7A9 and SLC3A1) across a panel of cancer cell lines. As shown in **rebuttal letter Fig. 6A, B** (Fig. 1a-b in the manuscript), cystine uptake levels in general correlated with expression levels of SLC7A11 (but not with those of SLC7A9 and SLC3A1) in these cell lines. We further categorized these cell lines into SLC7A11-low, -moderate, and -high cells based on their relative SLC7A11 expression and cystine uptake levels, in which SLC7A11-moderate (UMRC6, H226, A498, and A549) and -high cell lines (T98G and Hs578T) exhibited 2-5- and 10-fold increases, respectively, in cystine uptake compared to SLC7A11-low cell lines (H1299 and 786-O) (such that SLC7A11-low cells exhibited cystine uptake levels at $\leq 10 \times 10^3$ DPM, SLC7A11-high cells at $\geq 55 \times 10^3$ DPM, whereas SLC7A11-moderate cells had cystine uptake levels between 10×10^3 and 55×10^3 DPM).

Importantly, knocking-down of SLC7A11 in SLC7A11-moderate or -high cells resulted in different phenotypes in terms of H₂O₂-induced cell death: while knocking down SLC7A11 in SLC7A11-moderate cells increased H₂O₂-induced cell death, as expected (**rebuttal letter Fig. 6C-F**; Fig. S1b, c in the manuscript), knocking down SLC7A11 in SLC7A11-high cells actually suppressed H₂O₂-induced cell death (see Fig. 1c-j in the manuscript). This is consistent with our model that high and moderate expression levels of SLC7A11 have opposing effect on H₂O₂-induced cell death (see Discussion at pages 21-22 in our manuscript).

Figure 6. Protein levels of SLC7A11, SLC7A9 and SLC3A1 (A) and corresponding cystine uptake levels (B) in a panel of cancer cell lines. (C-F) SLC7A11 knock down sensitizes SLC7A11-moderate cell lines to H₂O₂-induced cell death. **: P<0.01; *: P<0.001.**

Conversely, we can also overexpress SLC7A11 in SLC7A11-low cells (H1299 and 786-O) at different levels to achieve moderate or high cystine uptake levels, resulting in suppression or promotion of H₂O₂-induced cell death (see Fig. 1k-n in the manuscript). However, it is important to note that we had to overexpress SLC7A11 in SLC7A11-low cells to a higher level than that of endogenous SLC7A11 in SLC7A11-moderate cell lines in order to achieve a corresponding moderate increase of cystine uptake. This is because SLC7A11 requires its partnering with the chaperone protein SLC3A2 in order to localize on the plasma membrane and

mediate cystine uptake, and SLC3A2 also binds to multiple other amino acid transporters in cells; consequently, we need to overexpress SLC7A11 at higher levels to compete with other transporters for partnering with certain amount of SLC3A2 (in other words, some of the overexpressed SLC7A11 are nonfunctional because they do not bind to SLC3A2). Therefore, quantifying SLC7A11 protein levels and cross-comparing SLC7A11 levels between cell lines with and without SLC7A11 overexpression can be misleading to readers; instead, in this case, comparing their cystine uptake levels would be more accurate. In addition, while cystine uptake levels in general correlated with SLC7A11 expression levels in these cell lines, the fold changes of cystine uptake (**Panel B**) were much more moderate than those of SLC7A11 levels (see the numbers under SLC7A11 blotting in **Panel A**) across SLC7A11-low/-moderate/-high cells, likely for similar reasons.

To summarize, (1) in cell lines without any genetic manipulation, definition of low-, moderate-, or high-expression of SLC7A11 can be guided by relative expression levels of endogenous SLC7A11 and corresponding cystine uptake rate in these cell lines; (2) for the reason listed above, it could be misleading to cross-compare SLC7A11 protein levels in cell lines with its overexpression vs those without overexpression. In this case, categorizing these cell lines based on their cystine uptake levels provide a more accurate and quantitative way to define these cell lines; (3) likewise, cross-comparing cystine and GSSG levels under H₂O₂ treatment between these cell lines can also be misleading, because cell death kinetics are different between these cell lines and consequently, we had to measure disulfide levels at different time points after H₂O₂ treatment (see also our response to question 6 from this reviewer).

3. It is necessary to consider whether gene overexpression and other cell construction methods may lose some expression during long-term cell culture, which will cause instability of the results.

We have frequently checked gene expression (such as SLC7A11 expression in SLC7A11-overexpressing or knockout cell lines) during this project, and made sure that gene expression levels were consistent throughout our analyses. Below, we addressed additional comments from the reviewer regarding instability issues of the results.

4. The influence of moderate overexpression of SLC7A11 either on cell death (fig1), intracellular cystine concentration (fig1a), or NADP⁺/NADPH ratio (fig4a), or on the growth and metastasis of the mouse tumor model (fig5a-d) have inconsistent changes in the high overexpression of SLC7A11, the results seems strange, the authors interpret in the discussion section said that, “We further propose that SLC7A11’s pro- or anti-cell death effect under H₂O₂ treatment is dictated by its expression level. Specifically, moderate overexpression of SLC7A11 in cancer cells appears to be beneficial, in that the antioxidant effect of GSH appears to be stronger than the NADPH-depleting effect of cystine reduction; consequently, moderate SLC7A11 overexpression suppresses H₂O₂-induced cell death (Fig. 6b).” But in fig2e, GSSG/GSH ratio seems have no change.

To facilitate our response to this question, we present the corresponding data in the manuscript as **rebuttal letter Fig. 7** here. Moderate overexpression of SLC7A11 (SLC7A11-moderate) resulted in increased cystine uptake (**panel A**) but without an obvious increase in intracellular

cystine levels (panel B; because cells cannot tolerate to build up toxic cystine to high levels and quickly reduce cystine to cysteine, as long as they have sufficient NADPH supply); consequently, SLC7A11 moderate overexpression increased intracellular cysteine (panel C) and glutathione (GSH) levels (panel D).

Figure 7. (A) Cystine uptake levels in SLC7A11-low, -moderate, and -high 786-O cells. (B-F) Measurement of intracellular concentrations of cystine (B), cysteine (C), GSH (D), GSSG (E) and the GSSG/GSH ratio (F) in SLC7A11-low, -moderate, and -high 786-O cells treated with vehicle or 1 mM H₂O₂ for 3.5 hours. (G) NADP⁺/NADPH ratio in H₂O₂-treated SLC7A11-low, -moderate, and -high 786-O cells. ****: P < 0.0001.

Increased GSH

levels would also promote its utilization in antioxidant defense, leading to increased GSSG levels, explaining moderately increased GSSG levels in cells with moderate overexpression of SLC7A11 (panel E). Because both GSH and GSSG levels are increased, there is no significant change in the GSSG/GSH ratio between SLC7A11-low and -moderate cells (panel F).

Perhaps this reviewer expected that SLC7A11 overexpression should decrease the GSSG/GSH ratio. However, under most normal cellular conditions (with a reducing intracellular environment), GSH concentration is much higher than GSSG concentration (compare panels D and E), and the GSSG/GSH ratio is very low (panel F). In our view, a further decrease in the GSSG/GSH ratio (which is already very low) would not mean much; instead, the more important readout for GSH function is the GSH concentration. Our results indeed showed that moderate overexpression of SLC7A11 increased GSH levels. Therefore, we believe that our description that “moderate SLC7A11 overexpression suppresses H₂O₂-induced cell death” because it can increase “the antioxidant effect of GSH” is accurate.

This pattern is drastically changed in cells with high overexpression of SLC7A11 (SLC7A11-high). SLC7A11-high cells had even higher cystine uptake levels and intracellular cysteine levels than SLC7A11-low cells (panels A, C). Under normal culture conditions (vehicle), GSH levels were not further increased from SLC7A11-moderate to -high cells (panel D), perhaps because GSH synthesis is already saturated in SLC7A11-moderate cells. Since cysteine itself and other cysteine-derived metabolites (such as taurine) also have antioxidant capability, we believe that under normal culture conditions SLC7A11-high cells still have stronger capability in antioxidant defense than SLC7A11-moderate cells. However, under H₂O₂ treatment, NADPH is depleted more rapidly in SLC7A11-high cells than in SLC7A11-low/-

moderate cells (**panel G**; apparent because of the combined NADPH depleting effect by H₂O₂ treatment and high rates of cystine reduction to cysteine). Under such a NADPH-depleting environment, cystine cannot be efficiently reduced to cysteine, leading to dramatic accumulation of intracellular cystine levels in SLC7A11-high cells (**panel B**); GSSG cannot be efficiently recycled back to GSH, leading to decreased GSH levels and drastically increased GSSG levels (and consequently, dramatically increased GSSG/GSH ratio) in SLC7A11-high cells than in SLC7A11-low/-moderate cells (**panels C-E**). We hope that our interpretation makes sense to this reviewer.

5. In fig1a-1d, low/moderate/high overexpression of SLC7A11 reduced cell death after administration of H₂O₂, high overexpression of SLC7A11 increased cell death, both moderate/high overexpression SLC7A11 increased cell death for glucose starvation. Cells with high endogenous expression of SLC7A11: knockout SLC7A11 increased cell death; knockdown SLC7A11 reduces cell death; for glucose starvation, both moderate/high overexpression SLC7A11 reduced cell death. But cysteine levels are not the cause of cell death; moreover the cause of cell death was not a normal one (like apoptosis, ferroptosis, etc.), and also not due to ROS (fig3&s3); what are the specific causes of cell death? And the relationship with glucose starvation and H₂O₂?

In another manuscript we recently published (5), we studied the nature of disulfide stress-induced cell death in the context of glucose-starved SLC7A11-moderate/-high cells. We found that aberrant accumulation of intracellular disulfides in SLC7A11-moderate/-high cells under glucose starvation induces a novel form of cell death distinct from apoptosis or ferroptosis. We termed this cell death disulfidptosis. Further functional studies revealed that glucose starvation in SLC7A11-moderate/-high cells induces aberrant disulfide bonds in actin cytoskeleton proteins, F-actin collapse, and cell contraction, which contributes to subsequent cell death (5).

H₂O₂-induced cell death in SLC7A11-high cells share several features with disulfidptosis (that is, glucose starvation-induced cell death in SLC7A11-moderate/-high cells), including (1) cells under both conditions exhibit aberrant levels of disulfide molecules (such as cystine); (2) cell death can be prevented by disulfide reducing agents (such as TCEP, 2-ME) or agents that regenerate free thiols via disulfide exchange (such as D- and L-penicillamine), indicating the cell death is caused by aberrant accumulation of disulfide molecules; and (3) cell death cannot be rescued by ROS scavenger (such as Trolox) or inhibitors that blocking other forms of cell death such as apoptosis and ferroptosis. We think H₂O₂-induced cell death in SLC7A11-high cells likely is disulfidptosis but further characterization is required. We added a discussion on this interesting question in the current manuscript (see colored text in pages 19-20), and hope this can stimulate further studies in the future.

6. In fig2d and 2i, the concentration of GSSG in 786-O cells with high overexpression of SLC7A11 after H₂O₂ treatment was much lower than that in T98G cells (with endogenous high expression of SLC7A11) after H₂O₂ treatment (1mM VS 6mM). Does it indicate that there are other factors besides SLC7A11 that play a key role in the concentration of disulfide metabolites such as GSSG?

We appreciate the question from the reviewer, but want to point out that it would be misleading to compare disulfide concentrations between different cell lines under H₂O₂ treatment. This is because different cell lines have different cell death kinetics upon H₂O₂ treatment, and correspondingly, we had to use different time points after H₂O₂ treatment for thiol measurement in these cell lines (time point information is provided in the corresponding figure legends). Specifically, we measured thiol concentrations in 786-O cells with high overexpression of SLC7A11 and T98G cells (with endogenous high expression of SLC7A11) at 3.5 hr and 6 hr after 1 mM H₂O₂ treatment, respectively. Of note, we chose the time points at which cells had not undergone obvious cell death (otherwise, it can be argued that whichever data we obtained merely reflects a secondary effect of cell demise). Because 786-O cells with high overexpression of SLC7A11 died more quickly than T98G cells under 1 mM H₂O₂ treatment (such that SLC7A11-overexpressing 786-O cells already undergo dramatic cell death under H₂O₂ treatment at 6 hr), we had to choose an earlier time point for thiol measurement in SLC7A11-overexpressing 786-O cells than in T98G cells, perhaps explaining the lower GSSG concentration in SLC7A11-overexpressing 786-O cells than in T98G cells.

Overall, our major point in these data is to show that H₂O₂ treatment can significantly increase disulfide levels in SLC7A11-high cells (either with endogenous high expression or with high overexpression), but these data are not intended for cross-comparison between different cell lines (because H₂O₂ treatment durations were different in these cell lines). We hope our interpretation makes sense to this reviewer.

Considering with follow-up questions, what concentration of GSSG, GSH and other metabolites is required to match the function and effect of endogenous high expression of SLC7A11?

Here we would like to limit our discussion to intracellular cystine concentration, because cystine is the direct substrate of SLC7A11 whereas others (such as GSSG, γ -glutamyl-cystine and γ -glutathionyl-cysteine) are downstream disulfide molecules derived from cystine. Based on our analyses under various disulfide stress-inducing conditions (glucose starvation or H₂O₂ treatment), a minimum of 0.7 mM cystine concentration appears to correlate with toxic levels of intracellular cystine for subsequent cell death induction (see Fig. 2a, h in this manuscript and **rebuttal letter Fig. 8**, which is derived from our previous publication (1)).

Figure 8. Cystine concentration in UMRC6 cells cultured with or without glucose.

Similar problems appear in fig3a-e, the concentration of cystine and GSSG is also ten times higher than that in fig2a and 2d. In the same group, how to explain the inconsistency of the quantitative concentration of metabolites such as cystine, GSSG?

As detailed below, we believe this reflects an inherent issue we have encountered during our analyses of this cell death rather than the inconsistency in our thiol measurement. In **rebuttal letter Fig. 9**, we present the comparison of cystine or GSSG peak area (after normalization to standard samples) from our thiol measurement, which shows comparable levels of cystine or GSSG between these two experiments shown in Figs. 2 and 3. Further conversion of peak area

data to concentration value requires normalization by PCV (packed cell volume). In these analyses, we used similar cell numbers, which should generate similar PCV values.

Figure 9. (A) Cystine peak area for Figure.2a and 3a. (B) Cystine concentration after normalization with PCV in Figure.2a and 3a. (C) GSSG peak area for Figure.2a and 3a. (D) GSSG concentration after normalization with PCV in Figure.2a and 3a.

However, we noticed that H₂O₂-induced cell death in SLC7A11-high cells is accompanied with dramatic cell contraction, resulting in decreased cell volume and therefore PCV value (even with similar cell numbers); furthermore, the cell contraction appears to be highly dynamic, such that in the experiment shown in Fig. 2 the PCV value is 1.6 for SLC7A11-high cells under H₂O₂ treatment, whereas in the experiment shown in Fig. 3 the PCV value became 0.3 with the same cell line and treatment condition (see the **table** below). As shown in **rebuttal letter Fig. 9**, the normalization with the PCV results in a significantly higher cystine or GSSG concentration in data shown in Fig. 3 than those in Fig. 2 (in other words, the significantly higher cystine or GSSG concentration in Fig. 3 reflects the decreased PCV value in these cells).

We now know what causes cell contraction in these cells. In another manuscript we recently published (5), we studied the nature of disulfide stress-induced cell death in the context of glucose-starved SLC7A11-moderate/-high cells. We found that aberrant accumulation of intracellular disulfides in SLC7A11-moderate/-high cells under glucose starvation induces a novel form of cell death distinct from apoptosis or ferroptosis. We termed this cell death disulfidptosis. Further analyses revealed that glucose starvation in SLC7A11-moderate/-high cells induces aberrant disulfide bonds in actin cytoskeleton proteins, F-actin collapse, and cell contraction, which contributes to subsequent cell death. We believe that H₂O₂-induced cell death in SLC7A11-high cells most likely is disulfidptosis (although further characterization is needed; see our response to question 5 from this reviewer), which explains the decreased PCV values in H₂O₂-treated SLC7A11-high cells. Unfortunately, the highly dynamic nature of cell contraction during H₂O₂ treatment adds inconsistency for our calculation of thiol concentrations between different experiments. We want to emphasize that this inconsistency does not affect our main conclusion. However, if this reviewer believes this remains a significant issue, we can instead present original peak area data for cystine and GSSG levels in our manuscript (as we showed in Fig. 1d for glutamyl-cystine).

For the experiment in Fig.2a and 2d the PCV for each sample are as follow:

	SLC7A11-Low	SLC7A11-Moderate	SLC7A11-High
Vehicle	2.0	2.0	1.8
H2O2	2.0	2.0	1.6

For the sample in Fig.3a-e, the PCV for each sample are as follow:

SLC7A11-Low	SLC7A11-High
-------------	--------------

Vehicle	H2O2	Vehicle	H2O2						
			DMSO	Erastin	TCEP	2-ME	NAC	D-Peni	L-Peni
2.0	2.0	1.5	0.3	1.5	2.0	1.5	1.5	2.0	1.75

7. From the results of fig4g-j, not enough evidence could support the conclusion of "Together, our data suggest that high expression of SLC7A11 in combination with H2O2 treatment depletes NADPH, which contributes to H2O2-induced cell death."

This reviewer argues that we do not have strong evidence to prove it is the NADPH depletion that causes H₂O₂-induced cell death in SLC7A11-high cells. Establishing this conclusion would require rescue data showing that restoring NADPH levels can suppress cell death in H₂O₂-treated SLC7A11-high cells. However, we cannot directly supplement NADPH into cells (because NADPH is highly unstable). In our previous study (1), we used 2DG treatment as an approach to supply NADPH in glucose-starved SLC7A11-moderate/-high cells. However, as discussed in the manuscript, we cannot use 2DG as an approach to supply NADPH here because under H₂O₂ treatment, cells were cultured in a glucose-replete medium (under which condition glucose continues to support NADPH generation). Considering this, we have toned down our conclusion here to "Together, our data suggest that high expression of SLC7A11 in combination with H₂O₂ treatment depletes NADPH, which potentially contributes to H₂O₂-induced cell death".

8. As it mentioned in the paper "In 786-O xenograft tumor models, SLC7A11-high tumors exhibited significantly increased tumor growth compared with SLC7A11-moderate or -low tumors (Fig. 5a), which was consistent with our and others' previous findings showing that SLC7A11 promotes tumor growth by suppressing ferroptosis", the results of figs 1e-f suggest that the use of ferroptosis inhibitors in 786-O and T98G has no effect on cell death in vitro, whether SLC7A11 overexpression or knockout, or with treatment of H2O2.

Fig. S1e-f show that H₂O₂-induced cell death in SLC7A11-high cells is not ferroptosis (because the ferroptosis inhibitor could not suppress this cell death). Since we did not study ferroptosis in this manuscript, to avoid confusion, we have changed the description in this sentence from "which was consistent with our and others' previous findings showing that SLC7A11 promotes tumor growth partly by suppressing ferroptosis" to "which was consistent with our and others' previous findings revealing a role of SLC7A11 in promoting tumor growth". This change will not affect the main conclusion of our study.

9. In this paper, only one breast cancer cell line were used in vitro, and tissue sample data of the Cancer Genome Atlas dataset and breast CTCs with unmatched control breast cancer tissue sample data were used to compare the expression level of SLC7A11. It seems not enough to say "Together, our data indicate that high SLC7A11 overexpression promotes primary tumor growth but suppresses metastasis, likely because SLC7A11-high cancer cells are susceptible to cell death induced by oxidative stress during metastasis. "

To address this comment from the reviewer, we now toned town our conclusion to "Together, our data indicate that, at least in the cell lines and tumor models we have examined, high SLC7A11 overexpression promotes primary tumor growth but suppresses metastasis, likely because SLC7A11-high cancer cells are susceptible to cell death induced by oxidative stress

during metastasis.” Of note, to address question 6 from reviewer 3, we have repeated our animal studies in another tumor model (H1299 model) and made similar observations (that is, high [but not moderate] overexpression of SLC7A11 promoted tumor growth yet suppressed metastasis; see **rebuttal letter Fig. 14A-E**).

Reviewer #3 (Remarks to the Author):

The paper is a follow up of a previous publication from this group examining the role of SLC7A11 in controlling cell death and survival. SLC7A11 imports cystine and has been shown to protect cells from excessive ROS and ferroptosis. However, reduction of cystine to cysteine requires NADPH and - as shown previously by this group - limiting glucose to high SLC7A11 expressing cells results in decreased NADPH production through the PPP and cell death due to disulfide accumulation. In the present paper the authors show that the level of SLC7A11 can determine the response to oxidative stress – with moderate expression protecting cells while high levels leading to cell death due to the accumulation of intracellular cystine.

The results presented largely follow a logical progression from the author's previous work. However, the physiological relevance of the work isn't quite clear.

1. The authors show that modulating the expression of SCL7A11 can result in changes of cystine uptake and increase or decrease of cell death. While the cell death associated with high SLC7A11 expression is shown not to be apoptosis or ferroptosis, it's not clear what the moderate expression of SLC7A11 is protecting from. Is this ferroptosis?

This reviewer asked an insightful question. Specifically, our data showed that moderate overexpression of SLC7A11 in SLC7A11-low cells (such as 786-O cells) suppressed H₂O₂-induced cell death. The reviewer asked the question: what is the nature of H₂O₂-induced cell death in SLC7A11-low cells? To address this question, we tested whether this cell death can be rescued by any cell death inhibitor. As shown in **Rebuttal letter Fig. 10A** (Fig. S11 in the revised manuscript), the apoptosis inhibitor Z-VAD and the necroptosis inhibitor Nec-1s, but not the ferroptosis inhibitor Fer-1, partially suppressed H₂O₂-induced cell death in SLC7A11-low cells, suggesting that H₂O₂ mainly induces apoptosis and necroptosis in SLC7A11-low cells. This conclusion is consistent with another recent publication (6) (**Rebuttal letter Fig. 10B**; data derived from (6)). These data together suggest that H₂O₂ induces different types of cell death in SLC7A11-low and -high cells.

Figure 10. H₂O₂ induces apoptosis and necroptosis in SLC7A11-low cells. (A) H₂O₂ induced cell death in SLC7A11-low cells can be partially rescued by the apoptosis inhibitor Z-VAD or the necrosis inhibitor (Nec-1s) in SLC7A11-low 786-O cells. (B) Decreases in cell viability induced by H₂O₂ can be partially rescued Z-VAD and Nec-1. **: P<0.01, ****: P<0.0001.

2. It seems that several cancer cell lines express high levels of SLC7A11 that support more cell death under both ROS and glucose starved conditions than seen in cells with moderately reduced SCL7A11. Can the authors speculate on why these tumour cells would be selected to carry such high expression – which seems to be generally detrimental to survival?

This reviewer asked an excellent question. Our previous data showed that cancer cells with moderate or high expression of SLC7A11 expression are susceptible to glucose starvation-induced cell death (1); that is, either moderate or high overexpression of SLC7A11 in SLC7A11-low cells promotes glucose starvation-induced cell death, whereas either complete KO or moderate knockdown of SLC7A11 in SLC7A11-high cells suppresses this cell death (see Fig. 1 in the current manuscript). Because under most conditions cancer cells or tumors are supplied with sufficient glucose, we believe that this metabolic vulnerability per se does not cause much issue in SLC7A11-moderate/-high tumors under normal conditions, and therefore does not select against tumors to carry high SLC7A11 expression.

Figure 11. Protein levels of SLC7A11, SLC7A9 and SLC3A1 (A) and corresponding cystine uptake levels (B) in a panel of cancer cell lines. (C-F) SLC7A11 knock down sensitizes SLC7A11-moderate cell lines to H₂O₂-induced cell death. **: P<0.01; ***: P<0.001.

However, the phenotypes in these cells under H₂O₂ treatment are different: moderate overexpression of SLC7A11 protects whereas high overexpression of SLC7A11 dramatically promotes H₂O₂-induced cell death. Because tumor cells, particularly metastasizing tumor cells, constantly experience oxidative stress in vivo, these data suggest that moderate overexpression of SLC7A11 should be most beneficial for tumor survival and metastasis and therefore should be selected for during tumor evolution. Our data indeed support this premise, as we found that cancer cells with very high expression of SLC7A11 (such as T98G and Hs578T) are relatively rare, and most cancer cells exhibit relatively moderate levels of SLC7A11 expression and cystine uptake, such as A498, UMRC6, H226, and A549 cells (**rebuttal letter Fig. 11A-B**; Fig. 1a-b in the manuscript). Importantly, while SLC7A11 knockdown even protected SLC7A11-high T98G and Hs578T cells from H₂O₂-induced cell death (see Fig. 1e-f, i-j in the manuscript), SLC7A11 knockdown in SLC7A11-moderate cells (A498, UMRC6, H226, and A549 cells) sensitized these cells to H₂O₂-induced cell death (**rebuttal letter Fig. 11C-F**; Fig. S1b-c in the manuscript). Together, our data suggest that too high or too low SLC7A11 levels could be detrimental to cancer cells, and tumors are selected to express moderate levels of SLC7A11.

3. In 786-O cells, moderate and high expression of SLC7A11 leads to a similar increase in cystine uptake (Fig 1e). However, there is a dramatic difference in the intracellular levels of cysteine and survival of these cells in response to H2O2 (Figure 2a). Does cystine uptake also show this strong increase in H2O2 treated cells? What is regulating this increase in cystine uptake in response to H2O2?

We would like to emphasize that the cystine uptake level in SLC7A11-high 786-O cells is indeed higher than that in SLC7A11-moderate counterparts (**rebuttal letter Fig. 12A**; Fig. 1m in the manuscript). Nevertheless, we agree that the increase of cystine uptake from SLC7A11-moderate to -high cells appears moderate, which contrasts with the more dramatic difference in cell death phenotypes in these two cell lines under H₂O₂ treatment.

Figure 12. H₂O₂ treatment increase cells' cystine uptake ability. (A) Cystine uptake levels in SLC7A11-low, -moderate, and -high 786-O cells. (B) Cystine uptake levels in SLC7A11-low, -moderate, and -high 786-O cells treated vehicle or H₂O₂. ****: P<0.0001.

To explain this discrepancy, the reviewer provided an insightful suggestion: since cell death was measured under H₂O₂ treatment, perhaps cystine uptake could also be modulated by H₂O₂ treatment, such that the difference in cystine uptake under H₂O₂ treatment in these cell lines might be more dramatic and therefore is more in line with cell death phenotypes in these cell lines. Our data indeed support the instinct from this reviewer: we found that H₂O₂ treatment increased cystine uptake in SLC7A11-low/-moderate/-high cells, but it seems that the increase in SLC7A11-high cells was more pronounced, such that there was more dramatic increase of

cystine uptake from SLC7A11-moderate to -high cells under H₂O₂ treatment (**rebuttal letter Fig. 12B**; Fig. S2c in the revised manuscript). Of note, another recent study also showed that H₂O₂ treatment can increase cystine uptake (7)); mechanistically, they showed that H₂O₂ treatment promotes cystine uptake at least partly by increasing cell surface expression of SLC7A11 (7).

Together, these data suggest that H₂O₂ treatment in SLC7A11-high cells not only consumes NADPH for H₂O₂ neutralization, but also promotes cystine uptake and its subsequent reduction to cysteine; this further drains NADPH reserves, leading to disulfide stress and subsequent cell death.

4. Can the authors rescue cell death by replenishing NADPH in these cells? Maybe activation of another NADPH generating system?

We agree that it would be nice to show that replenishing NADPH can suppress H₂O₂-induced cell death in SLC7A11-high cells. However, NADPH is unstable and cannot be used in direct supplementation. Our previous data showed that the pentose phosphate pathway (PPP) plays a major role in supplying NADPH for cystine reduction to cysteine; correspondingly, 2-deoxyglucose (2DG) treatment can supply NADPH through the PPP and suppress cell death in glucose-starved SLC7A11-overexpressing cells (1). However, as described in the current manuscript, we cannot use this approach as a rescuing approach for H₂O₂-induced cell death, because under H₂O₂ treatment, cells were cultured in a glucose-replete medium (under which condition glucose continues to support NADPH generation).

Figure 13. The effect of G6PD overexpression on H₂O₂-induced cell death in SLC7A11-high cells. (A) Western blot showing the overexpression of G6PD. (B) NADP⁺/NADPH ratio measurement in H₂O₂-treated SLC7A11-high cells overexpressing with empty vector or Myc-G6PD. (C) Cell death measurement in H₂O₂-treated SLC7A11-high cells overexpressing with empty vector or Myc-G6PD. ***: P<0.001; ns: not significant.

Likewise, we found that overexpressing G6PD (the rate-limiting enzyme in the PPP) did not significantly restore NADPH or suppress cell death in H₂O₂-treated SLC7A11-high cells (**rebuttal letter Fig. 13**), likely because overexpression of G6PD alone is not sufficient to significantly drive the PPP (as there are multiple components in the PPP), and/or because under glucose-replete condition, the PPP flux is already saturated. (Because this is a negative data and does not help establish our model, we only show the data in the rebuttal letter and do not include it in the manuscript.)

Despite of the lack of rescuing data, we hope our TPNOX data (which shows that depleting NADPH by TPNOX overexpression promoted H₂O₂-induced cell death in SLC7A11-

low cells) can still convince the reviewer of the causality between NADPH depletion and H₂O₂-induced cell death in this context. We have also toned down our conclusion on this part from “Together, our data suggest that high expression of SLC7A11 in combination with H₂O₂ treatment depletes NADPH, which contributes to H₂O₂-induced cell death.” to “Together, our data suggest that high expression of SLC7A11 in combination with H₂O₂ treatment depletes NADPH, which potentially contributes to H₂O₂-induced cell death.”.

5. High SLC7A11 expressing cells form much larger tumours in mice than the low or moderately expressing lines (Figure 5a). The authors suggest this is due to a protection from ferroptosis – I think it would be necessary to show that these cells are resistant to ferroptosis. What is inducing ferroptosis in these primary tumors, if not ROS or glucose depletion?

This question on Fig. 5a specifically refers to our following description in text “In 786-O xenograft tumor models, SLC7A11-high tumors exhibited significantly increased tumor growth compared with SLC7A11-moderate or -low tumors (Fig. 5a), which was consistent with our and others’ previous findings showing that SLC7A11 promotes tumor growth partly by suppressing ferroptosis (8-11).” The role of SLC7A11 in promoting tumor growth is well established, and this can be linked to its function in suppressing ferroptosis as well as other functions (we extensively discussed this in a recent review (10)). Since we did not study ferroptosis in this manuscript, we hope this reviewer will agree that further characterizing ferroptosis in these tumor models would be beyond the scope of this study. Correspondingly, we have changed the description from “which was consistent with our and others’ previous findings showing that SLC7A11 promotes tumor growth partly by suppressing ferroptosis” to “which was consistent with our and others’ previous findings revealing the role of SLC7A11 in promoting tumor growth”. This change will not affect the main conclusion of our study.

6. Given the dramatic difference in tumour growth it would be prudent to show the reproducibility of this with a different high SLC7A11 expressing line (H1299). Despite being significantly protected from ferroptosis, these cells are deficient in liver colonisation. Can this be rescued – for example with NAC but not with Trolox?

We agree it is important to extend our in vivo data to another cell line model. As shown in **rebuttal letter Fig. 14** (Fig. 5h-j, S5f-g in the manuscript), we showed that, in H1299 xenograft models, high (but not moderate) overexpression of SLC7A11 promoted tumor growth (**Panel A**) but suppressed metastasis (**Panels B-E**), which is consistent with our previous results with 786-O cells.

Furthermore, we showed that, in 786-O models, NAC treatment almost completely restored metastasis in SLC7A11-high tumors to the level similar to that in vehicle-treated SLC7A11-low tumors, whereas Trolox has a minimal rescuing effect on SLC7A11-tumors (**rebuttal letter Fig. 14F**; Fig. S5e in the manuscript), which is consistent with our in vitro data and suggests the cell death-promoting effect is at least partly responsible for high SLC7A11-mediated metastasis suppression.

Figure 14. High overexpression of SLC7A11 promotes primary tumor growth but suppresses tumor metastasis. (A) Measurement of tumor volumes in SLC7A11-low, -moderate, and -high H1299 xenograft tumors at different time points (days) after subcutaneous injection. (B) Images of bioluminescence in mice 30 min after intracardiac injection with SLC7A11-low, -moderate, and -high H1299 cells (left) and statistical analysis of whole-body photon flux (right). (C) Quantification of photon flux (photons per second) in mice normalized to day 0 after intracardiac injection of SLC7A11-low, -moderate, and -high H1299 cells (n = 7 mice for each group). (D) Images of bioluminescence in mice 3 weeks after intracardiac injection of SLC7A11-low, -moderate, and -high H1299 cells (left) and statistical analysis of the whole-body photon flux (right). (E) Representative images of liver metastasis from SLC7A11-low, -moderate, and -high H1299 cells. (F) Quantification of photon flux (photons per second) in mice normalized to day 0 after intracardiac injection of SLC7A11-low 786-O cells with PBS and -high 786-O cells treated with PBS, NAC, or Trolox (n = 4 or 5 mice for each group). **: P<0.01; ****: P<0.0001.

References:

1. Liu X, Olszewski K, Zhang Y, Lim EW, Shi J, Zhang X, Zhang J, Lee H, Koppula P, Lei G, Zhuang L, You MJ, Fang B, Li W, Metallo CM, Poyurovsky MV, Gan B. Cystine transporter regulation of pentose phosphate pathway dependency and disulfide stress exposes a targetable metabolic vulnerability in cancer. *Nat Cell Biol.* 2020;22(4):476-86 PubMed PMID: 32231310; PMCID: PMC7194135.
2. Ishii T, Mann GE. Redox status in mammalian cells and stem cells during culture in vitro: critical roles of Nrf2 and cystine transporter activity in the maintenance of redox balance. *Redox Biol.* 2014;2:786-94 PubMed PMID: 25009780; PMCID: PMC4085355.
3. Ishii T, Bannai S, Sugita Y. Mechanism of growth stimulation of L1210 cells by 2-mercaptoethanol in vitro. Role of the mixed disulfide of 2-mercaptoethanol and cysteine. *J Biol Chem.* 1981;256(23):12387-92 PubMed PMID: 7298664.
4. Sato H, Shiiya A, Kimata M, Maebara K, Tamba M, Sakakura Y, Makino N, Sugiyama F, Yagami K, Moriguchi T, Takahashi S, Bannai S. Redox imbalance in cystine/glutamate transporter-deficient mice. *J Biol Chem.* 2005;280(45):37423-9 PubMed PMID: 16144837.
5. Liu X, Nie L, Zhang Y, Yan Y, Wang C, Colic M, Olszewski K, Horbath A, Chen X, Lei G, Mao C, Wu S, Zhuang L, Poyurovsky MV, James You M, Hart T, Billadeau DD, Chen J, Gan B. Actin cytoskeleton vulnerability to disulfide stress mediates disulfidptosis. *Nat Cell Biol.* 2023 PubMed PMID: 36747082.
6. Jantas D, Chwastek J, Grygier B, Lason W. Neuroprotective Effects of Necrostatin-1 Against Oxidative Stress-Induced Cell Damage: an Involvement of Cathepsin D Inhibition. *Neurotox Res.* 2020;37(3):525-42 PubMed PMID: 31960265; PMCID: PMC7062871.
7. Chase LA, VerHeulen Kleyn M, Schiller N, King AG, Flores G, Engelsman SB, Bowles C, Smith SL, Robinson AE, Rothstein J. Hydrogen peroxide triggers an increase in cell surface expression of system x(c)(-) in cultured human glioma cells. *Neurochem Int.* 2020;134:104648 PubMed PMID: 31874187.
8. Jiang L, Kon N, Li T, Wang SJ, Su T, Hibshoosh H, Baer R, Gu W. Ferroptosis as a p53-mediated activity during tumour suppression. *Nature.* 2015;520(7545):57-62 PubMed PMID: 25799988; PMCID: 4455927.
9. Zhang Y, Shi J, Liu X, Feng L, Gong Z, Koppula P, Sirohi K, Li X, Wei Y, Lee H, Zhuang L, Chen G, Xiao ZD, Hung MC, Chen J, Huang P, Li W, Gan B. BAP1 links metabolic regulation of ferroptosis to tumour suppression. *Nat Cell Biol.* 2018;20(10):1181-92 PubMed PMID: 30202049; PMCID: PMC6170713.
10. Koppula P, Zhuang L, Gan B. Cystine transporter SLC7A11/xCT in cancer: ferroptosis, nutrient dependency, and cancer therapy. *Protein Cell.* 2021;12(8):599-620 PubMed PMID: 33000412; PMCID: PMC8310547.
11. Liu T, Jiang L, Tavana O, Gu W. The Deubiquitylase OTUB1 Mediates Ferroptosis via Stabilization of SLC7A11. *Cancer Res.* 2019;79(8):1913-24 PubMed PMID: 30709928; PMCID: PMC6467774.

REVIEWERS' COMMENTS

Reviewer #1 (Remarks to the Author):

The authors have addressed most of my concerns, and added new supportive evidence to demonstrate the role of SLC7a11 and NADPH in sensitivity to redox stress. I have no further comments.

Reviewer #2 (Remarks to the Author):

In the new Detailed point-by-point response to the reviewer's comments, the authors design much experiments to answer the questions, and tone down much inappropriate conclusions one by one, I think it partially solves the problem I raised, which could be accepted in its current state.

Reviewer #3 (Remarks to the Author):

In general, the authors have done a good job of revising the paper to address the reviewers' comments. I still have one major and a mor minor point outstanding.

1. My main concern is around the increased growth rate of SLC7A11 high tumors. I appreciate the effort the authors went to in presenting another in vivo model, which supports the original data well. However, I'm afraid I don't understand why SLC7A11 tumors grow much faster and I feel – for clarity – the authors need to address this issue, as it seems to be at odds with the main point of the paper, which is to define a vulnerability in SLC7A11 high tumors. It is possible that oxidative stress only kicks in during metastasis (although I think this is unlikely) – but this wouldn't explain why high levels of SLC7A11 are advantageous for the primary tumor. It's not clear whether the authors believe high levels of SLC7A11 protect from ferroptosis while promoting disulfidptosis? There is something interesting about the difference between primary tumor growth and metastasis that is not explained in the current version of the paper.

2. I asked why some cancers would develop with high levels of SLC7A11 and the authors replied that this is rare. But clearly it does occur, and I still wonder what the selective advantage of this might be, even in these rare tumours. Is this related to the faster growth rate of the SLC7A11 high cancers (see above)? I accept discovering the answer to this is beyond the scope of the study, but the authors should mention this (and any explanation they can think of) in the discussion.

RESPONSE TO REVIEWERS' COMMENTS

Reviewer #1 (Remarks to the Author):

The authors have addressed most of my concerns, and added new supportive evidence to demonstrate the role of SLC7a11 and NADPH in sensitivity to redox stress. I have no further comments.

We thank the reviewer for the support.

Reviewer #2 (Remarks to the Author):

In the new Detailed point-by-point response to the reviewer's comments, the authors design much experiments to answer the questions, and tone down much inappropriate conclusions one by one, I think it partially solves the problem I raised, which could be accepted in its current state.

We thank this reviewer for the support.

Reviewer #3 (Remarks to the Author):

In general, the authors have done a good job of revising the paper to address the reviewers' comments. I still have one major and a mor minor point outstanding.

We thank the reviewer for the support, and hope our following responses have addressed the remaining concerns from this reviewer.

1. My main concern is around the increased growth rate of SLC7A11 high tumors. I appreciate the effort the authors went to in presenting another in vivo model, which supports the original data well. However, I'm afraid I don't understand why SLC7A11 tumors grow much faster and I feel – for clarity – the authors need to address this issue, as it seems to be at odds with the main point of the paper, which is to define a vulnerability in SLC7A11 high tumors. It is possible that oxidative stress only kicks in during metastasis (although I think this is unlikely) – but this wouldn't explain why high levels of SLC7A11 are advantageous for the primary tumor. It's not clear whether the authors believe high levels of SLC7A11 protect from ferroptosis while promoting disulfidptosis? There is something interesting about the difference between primary tumor growth and metastasis that is not explained in the current version of the paper.

We appreciate the insightful question posed by the reviewer. SLC7A11 has a dual role in regulating cell survival and death: high expression of SLC7A11 protects cancer cells from ferroptosis but promotes disulfidptosis or H₂O₂-induced cell death (which we believe is likely disulfidptosis in SLC7A11-high cancer cells). The observed tumor-promoting effect by high overexpression of SLC7A11 is indeed expected and consistent with its established role in suppressing ferroptosis and apoptosis (both are tumor suppressive mechanisms) and in supporting cell proliferation. This is extensively discussed in our previous review on the role of SLC7A11 in cancer, as detailed in the sessions “SLC7A11 promotes tumor development partly via inhibiting ferroptosis” and “ferroptosis-independent functions of SLC7A11 in promoting tumor development” [1].

However, our current study showed an interesting result that high overexpression of SLC7A11 suppresses tumor metastasis, which may seem counterintuitive in light of its established tumor-promoting effect. We propose that this may be due to the excessive susceptibility of SLC7A11-high cancer cells to cell death induced by oxidative stress during metastasis. Notably, while primary cancer cells generally exhibit high oxidative stress (as correctly pointed out by this reviewer), metastasizing cancer cells often encounter even greater oxidative stress due to the hostile microenvironment they face as they break away from the primary tumor and travel to distant sites in the body [2]. Therefore, the susceptibility of SLC7A11-high cancer cells to oxidative stress-induced cell death may be more pronounced in metastasizing cancer cells than in primary tumor cells. This might help explain the differential effects of SLC7A11 high overexpression in primary tumor growth versus tumor metastasis. We hope our response provides further clarification on this question.

2. I asked why some cancers would develop with high levels of SLC7A11 and the authors replied that this is rare. But clearly it does occur, and I still wonder what the selective advantage of this might be, even in these rare tumours. Is this related to the faster growth rate of the SLC7A11 high cancers (see above)? I accept discovering the answer to this is beyond the scope of the study, but the authors should mention this (and any explanation they can think of) in the discussion.

We apologize for any lack of clarity in our previous response. Our current data suggest that SLC7A11-high cancer cells would be positively selected in primary tumors but negatively selected in metastasized tumors. Notably, completely ablating SLC7A11 expression also suppresses tumor metastasis [3], suggesting that moderate expression levels of SLC7A11 may be most beneficial for tumor metastasis. However, further investigations are required to fully test this hypothesis.

We have incorporated our points for both questions into our revised manuscript (see colored text in Discussion).

References:

1. Koppula, P., L. Zhuang, and B. Gan, *Cystine transporter SLC7A11/xCT in cancer: ferroptosis, nutrient dependency, and cancer therapy*. Protein Cell, 2021. **12**(8): p. 599-620.
2. Tasdogan, A., J.M. Ubellacker, and S.J. Morrison, *Redox Regulation in Cancer Cells during Metastasis*. Cancer Discov, 2021. **11**(11): p. 2682-2692.
3. Sato, M., et al., *Loss of the cystine/glutamate antiporter in melanoma abrogates tumor metastasis and markedly increases survival rates of mice*. Int J Cancer, 2020. **147**(11): p. 3224-3235.